# Neurophysiological evidence for crossmodal (face-name) person-identity representation in the human left ventral temporal cortex

Angélique Volfart[1,2], Jacques Jonas[1,3], Louis Maillard[1,3], Sophie Colnat-Coulbois[1,4], Bruno Rossion[1,2,3]*

1 Université de Lorraine, CNRS, CRAN, Nancy, France, 2 Université Catholique de Louvain, Institute of Research in Psychological Science, Institute of Neuroscience, Louvain-La-Neuve, Belgium, 3 Université de Lorraine, CHRU-Nancy, Service de Neurologie, Nancy, France, 4 Université de Lorraine, CHRU-Nancy, Service de Neurochirurgie, Nancy, France

* bruno.rossion@univ-lorraine.fr

**Data Availability Statement:** All EEG and SEEG data, together with a readme text file explaining the structure of the data, are available on a Dryad

## Abstract

Putting a name to a face is a highly common activity in our daily life that greatly enriches social interactions. Although this specific person–identity association becomes automatic with learning, it remains difficult and can easily be disrupted in normal circumstances or neurological conditions. To shed light on the neural basis of this important and yet poorly understood association between different input modalities in the human brain, we designed a crossmodal frequency-tagging paradigm coupled to brain activity recording via scalp and intracerebral electroencephalography. In Experiment 1, 12 participants were presented with variable pictures of faces and written names of a single famous identity at a 4-Hz frequency rate while performing an orthogonal task. Every 7 items, another famous identity appeared, either as a face or a name. Robust electrophysiological responses were found exactly at the frequency of identity change (i.e., 4 Hz / 7 = 0.571 Hz), suggesting a crossmodal neural response to person identity. In Experiment 2 with twenty participants, two control conditions with periodic changes of identity for faces or names only were added to estimate the contribution of unimodal neural activity to the putative crossmodal face-name responses. About 30% of the response occurring at the frequency of crossmodal identity change over the left occipito-temporal cortex could not be accounted for by the linear sum of unimodal responses. Finally, intracerebral recordings in the left ventral anterior temporal lobe (ATL) in 7 epileptic patients tested with this paradigm revealed a small number of "pure" crossmodal responses, i.e., with no response to changes of identity for faces or names only. Altogether, these observations provide evidence for integration of verbal and nonverbal person identity-specific information in the human brain, highlighting the contribution of the left ventral ATL in the automatic retrieval of face-name identity associations.

repository: https://doi.org/10.5061/dryad.m8t391m.

**Funding:** This study was supported by a LUE grant (http://lue.univ-lorraine.fr/en) and an FNRS/FWO grant (HUMVISCAT), grant no. 30991544 (https://www.frs-fnrs.be/fr/financements/credits-et-projets/eos). AV was supported by a grant of the Université de Lorraine (MENESR doctoral grant). The funders had no role in study design, data collection and analysis, decision to publish, or preparation of the manuscript.

**Competing interests:** The authors have declared that no competing interests exist.

**Abbreviations:** antCoS, anterior segment of the collateral sulcus; antFG, anterior fusiform gyrus; antMTG/ITG, anterior part of the inferior and middle temporal gyri; antOTS, anterior segment of the occipito-temporal sulcus; antPHG, anterior segment of the parahippocampal gyrus; ATL, anterior temporal lobe; CMS, common mode sense; DRL, driven right leg; EEG, electroencephalography; EOG, electrooculogram; FFT, fast Fourier transform; fMRI, functional magnetic resonance imaging; FPVS, fast periodic visual stimulation; LOT, left occipito-temporal region of interest; RT, response time; SD, semantic dementia; MTL, medial temporal lobe; PTL, posterior temporal lobe; ROI, region of interest; ROT, right occipito-temporal region of interest; SEEG, stereo electroencephalography; SSVEP, steady-state visual evoked potential; TMS, transcranial magnetic stimulation; VOTC, ventral occipito-temporal cortex.

## Introduction

Putting a name to a familiar face is a highly common activity in our daily life, which greatly enriches social interactions. For neurotypical adult individuals, this association is often automatic—i.e., names are retrieved even without the intention to do so—and yet quite difficult: often, someone's name cannot be remembered, or can take a few seconds to be evoked from one's face [1–3]. These difficulties increase with ageing, with people being often concerned with a reduced ability to retrieve specific names associated with familiar faces [4–7]. An inability to accurately put a name on a face is particularly salient in patients with Alzheimer disease [8,9], semantic dementia (SD) [10,11], or traumatic brain injury [12,13].

Many studies have described brain-damaged patients with an impaired ability to retrieve a name from a person's face with a preserved ability to provide semantic information about the person (e.g., the person's profession) [14–19]. The reverse pattern of impairment, i.e., impaired access to semantic information with preserved naming of a face, has not been described, to our knowledge. These findings have been interpreted within cognitive models of person recognition, in which person-related semantic representations act as a gateway to name retrieval [20–23].

The neural basis of face-name association remains largely unknown. Rare studies performed in epileptic patients with depth electrodes implanted in the medial temporal lobe (MTL) have revealed single neurons firing selectively to a famous person identity irrespective of the presentation format (e.g., Jennifer Aniston's face and her written name) [24]. However, while these neurons have been defined as "concept cells", they are thought to play a role in creation of associations and recollection of (recent or recently refreshed) conscious episodic memory events rather than supporting long-term semantic associations [25].

At the system level of brain organization, two opposite views can be advanced. On the one hand, verbal and nonverbal person-specific information may be processed separately in the left and right anterior temporal lobe (ATL), respectively, and integrated through re-entrant interhemispheric connections [26,27]. This proposal relies on the consistent observation that the left and right cerebral hemispheres contribute differently to the processing of names and faces, respectively. In neuroimaging, familiar faces activate most strongly the right occipito-temporal cortex, whereas written proper names preferentially activate the left hemisphere [28–32]. Consistent with these studies, group studies of SD patients have shown that the recognition of famous names is relatively more affected by a left anterior temporal atrophy, while famous face recognition deficits are relatively more affected by a right anterior temporal atrophy [11,33,34]. Neuropsychological single-case studies of patients with anterior temporal atrophy [35–38] or focal anterior temporal damage [39] have also found evidence for graded hemispheric dissociations within person-related recognition abilities, i.e., relative differential impairments in accessing person-related information from faces or names depending on the hemispheric side of the lesion. Moreover, the lack of crossmodal adaptation effects to sequences of faces and names with the same identity in functional magnetic resonance imaging (fMRI) [40] has reinforced this view of two separate processing pathways for faces and names in the human brain.

On the other hand, other authors have proposed a semantic "hub" in the bilateral ATLs, integrating modality-specific representations into a shared amodal/transmodal representation [41–45]. This theory was first grounded in the observation of SD patients whose progressive bilateral ATL atrophy has been systematically associated with a severe multimodal semantic impairment [46–48]. According to this view, modality-specific person-related representations stored in different parts of the brain ("spokes") are integrated into a common individual representation in the ATL (hub). Within this bilateral semantic network, between-hemisphere

differences are relative rather than absolute and can be interpreted as a consequence of the differential connectivity with other regions such as language areas and the left hemisphere [49,50]. This "hub-and-spoke" model of semantic processing in the human brain has been supported by a wealth of evidence ranging from clinical observations [41,46,48,51], functional neuroimaging [45,52–55], transcranial magnetic stimulation (TMS) [56,57], and computational simulations [41,58]. Regarding the specific association of familiar faces and names, which is only a subcomponent of person semantics, the hub-and-spoke model predicts modality-specific representations of a person's identity ("spokes"), in addition to an integrated multimodal representation in the bilateral ATL. Supporting this view, neuroimaging studies have provided evidence that the recognition of famous faces and names activates the same regions in the bilateral temporal lobes [28,45,59,60], with potential re-entrant interactions to the cortical face network leading to priming effects between faces and names of the same identity [61,62]. However, the lateralization of these common person-related representations appears inconsistent across studies, i.e., being either unilateral [28,60] or bilateral [45,59,63], possibly because these studies have often used explicit naming tasks, which involve different components of language processing and are thus strongly left lateralized. Most importantly, irrespective of the hemispheric lateralization issue, to our knowledge, no direct evidence has been provided so far for an integrated neural representation of faces and names at the level of a specific person identity. That is, beyond evidence for neural populations responding to both faces and names in addition to unimodal responses, what is critically lacking at present is experimental evidence for neural integration of a specific face and a name identity into a common, i.e., crossmodal, representation.

To provide evidence for such a crossmodal neural representation of person identity, we developed an original crossmodal frequency-tagging paradigm coupled to brain activity recording via both scalp and intracerebral electroencephalography (EEG). Our approach is based on the early observation that the brain's electroencephalographic activity synchronizes exactly with the temporal frequency of a periodic visual stimulus (e.g., a flickering light at 17 Hz generates an EEG response at 17 Hz) [64]. These digitally captured periodic EEG responses, often named steady-state visual evoked potentials (SSVEPs), can be expressed in the frequency domain through Fourier transform [65,66]. Compared to standard recordings of event-related potentials, this approach has substantial advantages in terms of sensitivity (high signal-to-noise ratio) and objectivity (i.e., the response occurs at a frequency predefined by the experimenter and can be quantified easily) [67]. Over the last decade, this "frequency-tagging" or "fast periodic visual stimulation" (FPVS) approach in human EEG has been successfully extended to study higher-level visual functions, in particular using periodic "oddball" stimulation to measure face [68,69] and written words [70] categorization. The term "SSVEP" is not used here because it refers to the type of response obtained rather than the approach, and is a loaded term, with different researchers having different views on what is and what is not a SSVEP (as opposed to a transient event-related potential) [71]. The term "frequency-tagging" is more appropriate, although often used in the context of spatially distinct stimuli flickering at different frequencies [67,72], and this is why the term "FPVS" is preferred here.

Based on these principles and observations, in the present study, we performed a first experiment in which variable pictures of faces and written names of a single famous identity were presented at 4 Hz (i.e., 250-ms stimulus onset asynchrony) while recording brain activity in healthy volunteers with scalp EEG. At every stimulation cycle, either a picture or a name of that same ("base") identity appeared, with the items selected randomly in a large pool of variable pictures and names (Fig 1). Hence, compared to previous studies, the FPVS approach is extended here to a multimodal (i.e., faces and written names) stimulation. Importantly, participants in the study did not have to explicitly associate the names and faces, only perform an

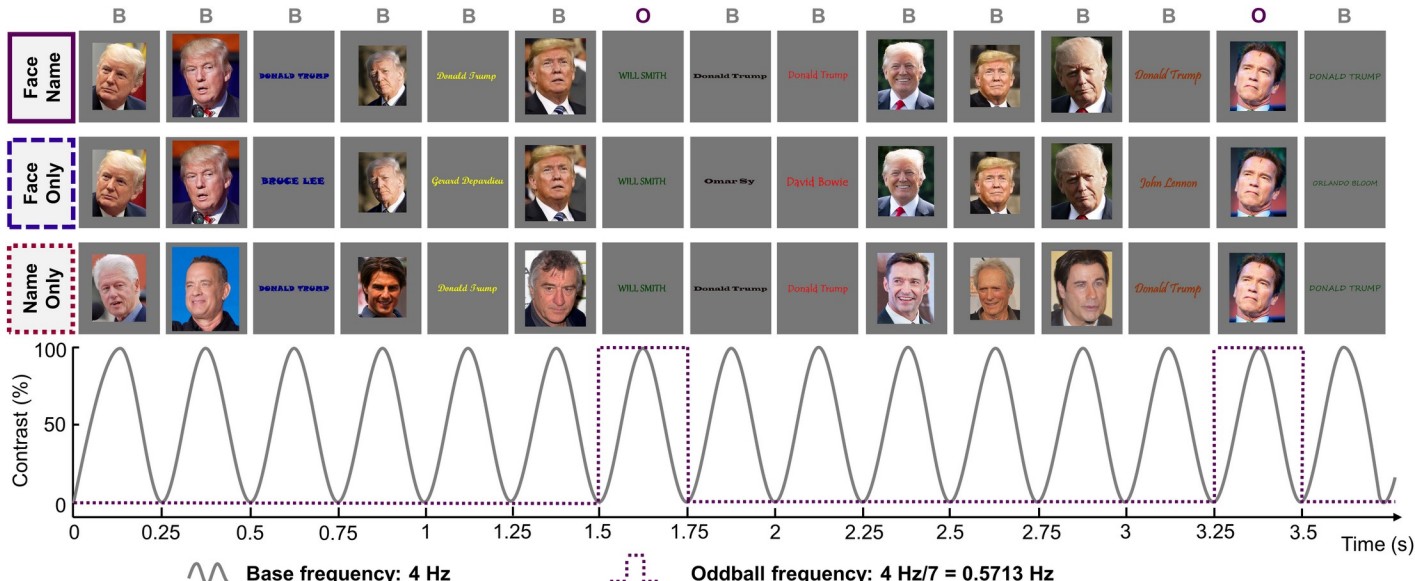

**Fig 1. Design of Experiments 1 and 2.** Example of a sequence with the identity D. Trump as base stimuli. Experiment 1 included the condition Face Name only (top row), with four famous identities as frequent stimuli. Experiment 2 used two famous identities, but included all 3 conditions: Face Name, Face Only, and Name Only. Visual stimuli (names and faces) are presented at a frequency of 4 Hz with a sinusoidal contrast modulation, and an identity change occurs every seven stimuli (every 0.571 Hz = 4 Hz / 7). For viewing purpose, faces and names are showed at the same position for the three conditions, but the actual order of stimuli was randomized. In Experiment 1, name stimuli were only uppercase and written in more conventional fonts (e.g., Arial, Agency, etc.) than those displayed in the figure. In Experiment 2, name stimuli could be either uppercase or lowercase and were written in less conventional fonts to increase the variability among the stimuli and reduce the potential reliance of the response on low-level visual confounds. Note that face images shown here are not the exact same images as in original paradigms but equivalent images with copyright agreements (CC-BY licenses). Sources (by order of appearance in the figure, left to right, top to bottom) are as follows: https://flic.kr/p/2bnDkJ6; https://flic.kr/p/Mj9V9J; https://flic.kr/p/MBK3za; https://flic.kr/p/MBK4u6; https://flic.kr/p/PeJVCb; https://flic.kr/p/PeJX47; https://flic.kr/p/PeJXHy; https://flic.kr/p/aHvjQc; https://flic.kr/p/NeRpv8; https://flic.kr/p/Ms12cD; https://flic.kr/p/8kTTLh; https://flic.kr/p/7Ybc2J; https://flic.kr/p/DwpstN; https://flic.kr/p/mAWhLD; https://flic.kr/p/nxmqZi.

unrelated orthogonal task throughout the stimulation. Every seven items, a different famous identity appeared, either as a face or as a name. If the face and name of the famous base identity are automatically associated during this stimulation sequence, then the occurrence of a different identity among the flow of same-identity stimuli should disrupt this association. Critically, a population of neurons sensitive to person identity from both modalities (written names and face pictures) should therefore adapt/habituate at every stimulation cycle when the given identity is repeated, irrespective of the change of stimulation format and (most importantly) stimulation modality. In this case, identity-oddball responses should emerge in the EEG whether faces or names interrupt this repetition at the periodic "identity-oddball" rate (Fig 1, top row). Consequently, if faces and names of a given identity are automatically integrated into a common crossmodal semantic representation, an electrophysiological identity-oddball response should appear in the EEG spectrum exactly at the frequency of person identity change (4 Hz / 7 = 0.571 Hz), even without any explicit face- or name-related task from our participants.

In a second experiment, having established the presence of neural responses reflecting the putative integration of identity-related faces and names, we ensured that these responses were not due to the mere sum of independent populations of neurons showing identity-oddball responses because of the statistical regularity of the modal identity-oddball response (i.e., a population of face-selective neurons responding to identity-oddball faces falling more often every 7 stimuli than any other regularity in the sequence; e.g., for faces: f1-n1-n1-f1-f1-n1-_f2_-f1-n1-f1-n1-n1-f1-_f3_-n1-n1, etc.). To do that, we added two control conditions to the Face

Name condition (FN), investigating the potential contribution of modality-specific processes to the face-name association response (Fig 1). In the Face Only condition (FO), names of the base identity were replaced by other famous names in order to isolate unimodal face responses that potentially contribute to the neural response observed in the Face Name condition. In the Name Only condition (NO), face images of the base identity were replaced by other famous faces, isolating unimodal name responses potentially contributing to the face-name response. Importantly, a different famous face or name identity was always presented every 7 items in all conditions, and pictures of faces and written names appeared as frequently in all conditions. However, in the two control conditions, automatic association of faces and names of a specific identity was not possible.

Two alternative hypotheses were considered: (1) If face-name association results from the conjoint activation of separated brain regions containing modality-specific person-related representation ("face" regions and "name" regions), then electrophysiological activity in the Face Name condition should not differ from the sum of the electrophysiological responses in the Face Only and Name Only conditions (additive processing; FN = FO + NO). (2) If, on the contrary, a face-name association also emerges from regions coding shared person-related representations, then the electrophysiological response in the Face Name condition should be greater than the sum of the activities in the two control conditions (integrative processing; FN > FO + NO). Given the above debate, the lateralization of the putative specific crossmodal response evoked in the Face Name condition was also of high interest.

Finally, we had the rare opportunity to test this paradigm in seven epileptic patients implanted in the left ventral ATL with depth electrodes (stereo electroencephalography or SEEG), in order to search more directly for local integrated face-name representations, i.e., face-name association responses without modality-specific responses.

## Results

As for materials and methods, this section will be divided into three parts: Experiment 1, Experiment 2a in scalp EEG, and Experiment 2b in intracerebral EEG. Our first hypothesis was that face and name representations of the famous base identity are automatically integrated during the Face Name sequence. Therefore, we should observe electrophysiological oddball responses at the frequency of person identity change (4 Hz / 7, i.e., 0.571 Hz) and its harmonics.

### Experiment 1

**Face-name responses.** Consistent with our hypothesis and despite randomly mixing faces and written names (Fig 1), we found clear responses at the frequency of person identity change (4 Hz / 7, i.e., 0.571 Hz and its harmonics). When considering the average of activity across the whole scalp, we found significant oddball responses at harmonics 3, 4, 5, and 6 (1.714 Hz, 2.285 Hz, 2.857 Hz, and 3.428 Hz; $p < 0.01$ for the third harmonic, $p < 0.05$ for the fourth harmonic, and $p < 0.001$ for the fifth and sixth harmonics).

Face-name responses were found over bilateral low occipito-temporal sites, below electrodes PO7 and PO8 (Fig 2). According to this topographical distribution, we defined two bilateral symmetrical occipito-temporal regions of interest (ROIs): electrodes P9, PO9, and PO11 for the left hemisphere (= left occipito-temporal [LOT]); P10, PO10, and PO12 for the right hemisphere (= right occipito-temporal [ROT]); and the middle occipital ROI lying in between (Oz, OIz and Iz), and where responses to the base rate were prominent (Fig 3). When considering only responses on pooled electrodes in the left and right ROIs (average of

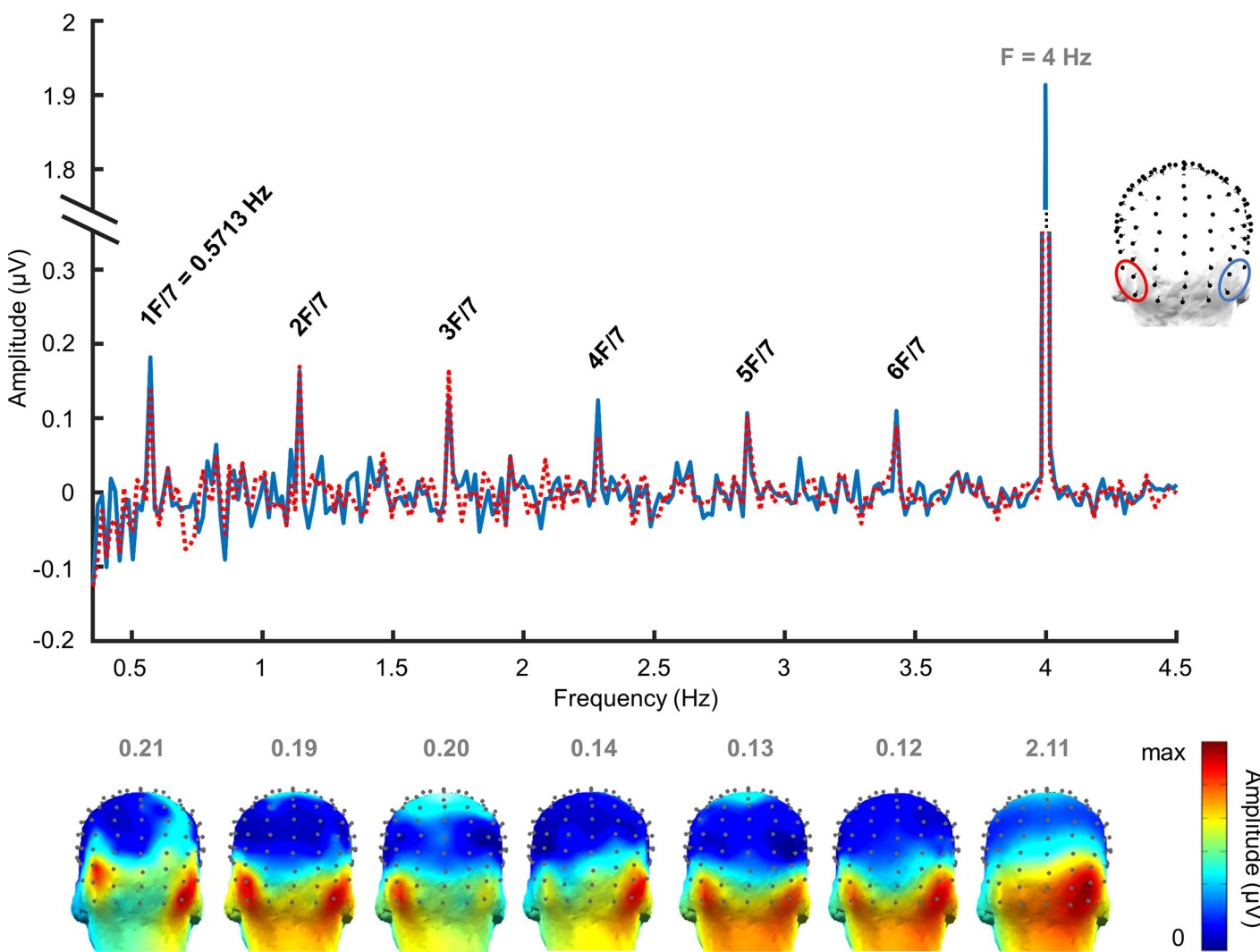

**Fig 2. FFT spectrum of EEG responses in Experiment 1.** The baseline-corrected FFT spectrum (in μV) in the left and right ROIs (mean amplitude of pooled electrodes) is presented (grand-averaged data, $n$ = 12 participants). The location of the ROIs is indicated by the topographical head on the right of the EEG spectrum. Black labels on the FFT spectrum refer to identity-oddball frequencies. All identity-oddball harmonics are significant at $p < 0.05$. The light gray label indicates the base frequency. Below the spectrum, 3D topographical distribution maps of responses are displayed for each identity-oddball harmonic and for the base frequency. Color scales' maxima are shown above each map and indicate the maximal baseline-corrected amplitude in μV at each harmonic of interest. Data underlying this figure are deposited on a Dryad repository: https://doi.org/10.5061/dryad.m8t391m. EEG, electroencephalography; FFT, fast Fourier transform; ROI, region of interest.

responses on the 3 electrodes in each ROI), all identity-oddball harmonics were significant in both hemispheres ($p < 0.05$) (Fig 2).

To quantify the periodic response related to person identity change, we computed the sum of six oddball harmonics (0.571 Hz, 1.714 Hz, 2.285 Hz, and so forth) regardless of base identity. We found a significant difference in baseline-corrected amplitudes (μV) between the three ROIs (F(2,22) = 13.87, $p = 0.0001$, $d = 2.26$). Whereas no difference was found between face–name responses in the LOT and ROT regions (1.09 μV ± 0.48 and 1.12 μV ± 0.49, respectively; $p = 1$), the amplitude of responses in the middle ROI (0.654 μV ± 0.40) was significantly lower than in occipito-temporal ROIs ($p = 0.0003$ for the difference with the ROT; $p = 0.008$ with the LOT). This effect was not due to variations of the noise level between the middle

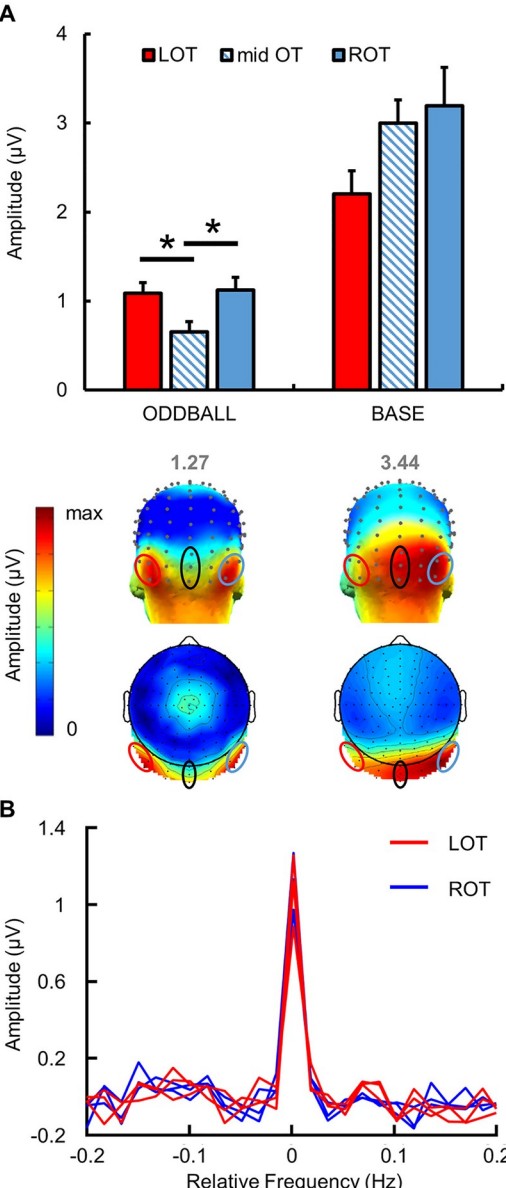

**Fig 3. Quantification and scalp topography of EEG responses in Experiment 1.** (A) Mean identity-oddball and base responses in μV (baseline-corrected amplitude) at the sum of 6 harmonics for the three ROIs (3 electrodes each) (grand-averaged data, *n* = 12 participants). Error bars indicate standard error of the mean, reflecting variability across participants. Asterisks indicate significant differences with a *p*-value < 0.05. Below, 3D and 2D topographies of oddball and base responses at the group level. Color scales' maxima are shown above each map, corresponding to the maximal baseline-corrected amplitude value. (B) Chunked segmentation centered on the oddball frequency and summed across six harmonics, with LOT channels in red and ROT channels in blue. Data underlying this figure are deposited on a Dryad repository: https://doi.org/10.5061/dryad.m8t391m. LOT, left occipito-temporal; mid OT, middle occipito-temporal; ROT, right occipito-temporal.

occipital ROI and the lateral ROIs: the standard deviation of the noise (standard deviation of amplitudes in the bins surrounding the identity-oddball frequency, i.e., 22 bins) at the electrodes of each ROI was not different between the lateral (left and right) ROIs and the middle occipital ROI ($F_{(2,6)} = 0.935$, $p = 0.443$) (S1 Fig).

Overall, these results confirm the predominance of bilateral occipito-temporal responses at the frequencies of interest when considering the whole group (Fig 3A and 3B).

Impressively, at the individual level, significant responses were found in 100% of participants for the sum of harmonics across all electrodes of the bilateral ROIs and across identities (Z-score range across subjects = 2.59–11.60; all $p$-values < 0.01) (S2 Fig).

**Base frequency responses.** A general visual response was expected at 4 Hz, frequency of the base stimulation, thought to reflect general visual synchronization processes. Consistently, there were clear significant base responses ($p < 0.001$) up to the sixth base harmonic (6F = 23.994 Hz) when considering the average of all electrodes. The topographical scalp distribution of base frequency responses showed an overall distribution over the right middle occipital region (Fig 3A).

To quantify base stimulation responses, we computed the sum of the first six base harmonics and contrasted baseline-corrected amplitudes (in µV) in each ROI. A significant difference was found between ROIs (F(2,22) = 4.92, $p = 0.017$, $d = 1.34$). Post hoc analysis showed a trend towards statistical difference between the LOT (2.20 µV ± 0.26) and middle ROI (2.99 µV ± 0.26) ($p = 0.059$) and between the LOT and the ROT (3.19 µV ± 0.43) ($p = 0.065$) but no difference between the middle ROI and the ROT ($p = 1$).

## Experiment 2a (scalp EEG)

Experiment 1 showed a robust (face-name) oddball response over occipito-temporal sites when a change of identity occurred in the sequence, irrespective of the presentation modality of the base rate and oddball stimuli (i.e., an amodal oddball response). However, we cannot fully exclude that an oddball is generated due to the statistical irregularity in a single modality only, in different populations of neurons coding for facial identity of names and faces separately. If this is the case, the oddball response identified in Experiment 1 should merely be equal to the sum of the two modality-specific violations. Hence, in Experiment 2, we sought to replicate the results of Experiment 1 in the Face Name condition, and added two control conditions (Face Only and Name Only, Fig 1) to be able to determine whether face-name association responses were due to a genuine integrative process (i.e., FN > FO + NO). In this case, we sought to quantify the specific contribution of the integrated response to the overall identity-oddball response. Twenty participants were presented with these three conditions.

First, results in scalp EEG were grand-averaged across subjects, conditions and electrodes to determine the range of significant harmonic frequencies. The first six oddball harmonics (0.571 Hz, 1.714 Hz, and so forth; $p < 0.001$ for the second to the sixth harmonic) and the first nine consecutive base harmonics (4 Hz, 7.998 Hz, and so forth; all $p$-values < 0.001) were taken into consideration for further analyses.

**Comparison between Face Name and control conditions.** Grand-averaged data across subjects and electrodes were considered separately for the three conditions. In the Face Name condition, clear oddball responses were found at the second, third, fourth, fifth, and sixth harmonics ($p < 0.05$ for the second harmonic; $p$-values < 0.001 for the third to fifth harmonics; $p < 0.01$ for the sixth harmonic). In the Face Only condition, only the fifth and sixth oddball harmonics reached significance (respectively, $p < 0.01$ and $p < 0.05$), while in the Name Only condition, no oddball harmonic reached significance. These results show a greater proportion of significant oddball harmonic responses in the Face Name condition relative to the control conditions. Consistently, a significant difference was found between the three conditions at the sum of harmonics ($\chi^2(2) = 10.80$, $p = 0.005$, $d = 2.17$). Post hoc tests demonstrated a larger amplitude at the sum of oddball frequencies in the Face Name condition (0.32 µV ± 0.25) compared to Face Only (0.12 µV ± 0.10, $p = 0.003$) and Name Only (0.09 µV ± 0.13, p = 0.002) conditions. No significant difference was found between Face Only and Name Only conditions ($p = 0.39$).

As in Experiment 1, the Face Name condition elicited strong bilateral occipito-temporal responses. The topographical distribution of control conditions was different in that Face Only responses appeared to be strongly right lateralized, while Name Only responses exhibited a bilateral occipito-temporal activation (Fig 4A and 4B). According to these scalp topographies and considering that there was no significant response in the middle occipital ROI of Experiment 1, two bilateral occipito-temporal ROIs were defined: I1, POI1, PO11, PO9 and P9 for the left hemisphere (= LOT); and I2, POI2, PO12, PO10 and P10 for the right hemisphere (= ROT).

When considering only responses on pooled electrodes in the left and right ROIs (average of responses on the 5 electrodes in each ROI), the first to sixth oddball harmonics were significant in both hemispheres (all $p$-values $< 0.001$ except the first harmonic in both ROIs, $p < 0.01$) in the Face Name condition. In the Face Only condition, the second to sixth oddball harmonics were significant in both hemispheres (all $p$-values $< 0.001$ except the second harmonic in the left ROI, $p < 0.05$). In the Name Only condition, the third to sixth harmonics were significant in both hemispheres (all $p$-values $< 0.05$) (S3 Fig).

To quantify the periodic response, baseline-corrected amplitudes at the sum of harmonics were analysed by regions of interest. A two-way ANOVA with repeated measures showed a main effect of condition (F(2,38) = 5.996, $p < 0.001$, $d = 2.27$) but no main effect of ROIs (F(1,19) = 1.026, $p = 0.324$, $d = 0.46$). Post hoc tests showed significantly larger amplitudes in the Face Name condition compared to the Face Only and Name Only conditions ($p < 0.001$ for both comparisons) but no significant difference between the two control conditions ($p = 0.935$). A significant interaction was also found between ROIs and conditions (F(2,38) = 3.533, $p = 0.039$, $d = 0.86$), and post hoc tests showed a significant difference between the left and right ROIs in the Face Only condition ($p = 0.028$) but not in the two other conditions ($p$-values $> 0.05$).

These results replicated the findings of Experiment 1, with a bilateral occipito-temporal distribution of person identity responses, and highlighted a strong right lateralization of oddball responses in the Face Only condition. Moreover, the level of noise recorded on the electrodes of the two ROIs did not differ in all 3 conditions ($p$-value range = 0.30 to 0.98; S1 Fig).

**Comparison between Face Name condition and the sum of Face Only and Name Only.** Following our hypotheses on the neural organization of face-name association processes, we asked whether the response observed in the Face Name condition could result from the mere addition of the responses in Face Only and Name Only conditions. To answer this question, the uncorrected-baseline amplitudes of the two control conditions were summed at the sum of oddball harmonics across electrodes and then corrected (sum FO + NO). The topography of the sum (FO + NO) appeared to be right lateralized, compared to the bilateral distribution of face-name responses (Fig 5A). However, no significant difference was found between the ROIs at the sum (FO + NO) ($t = -1.208$, $p = 0.242$, $d = -0.294$) (no difference in noise level between left and right ROIs: $p = 0.135$). To highlight the difference in electrophysiological responses between Face Name and the sum (FO + NO), we also subtracted uncorrected-baseline amplitudes of the sum from uncorrected-baseline amplitudes of the Face Name condition [FN–(FO + NO)] and applied a baseline correction. This subtraction showed a left-lateralized selective response for the Face Name condition (Fig 5A).

To quantify this response, corrected-baseline amplitudes of the sum (FO + NO) were compared with amplitudes of the Face Name condition (Fig 5B). A two-way ANOVA with repeated measures showed a main effect of condition (F(1,19) = 5.003, $p = 0.037$, $d = 1.02$), revealing a greater amplitude of the effect at the oddball-identity frequency in the Face Name condition compared to the sum (FO + NO). There was no main effect of ROIs (F(1,19) = 1.031, $p = 0.323$, $d = 0.46$) and no interaction effect between ROIs and conditions (F(1,19) = 1.247, $p = 0.278$, $d = 0.51$). Further analyses (paired $t$ tests) showed that the

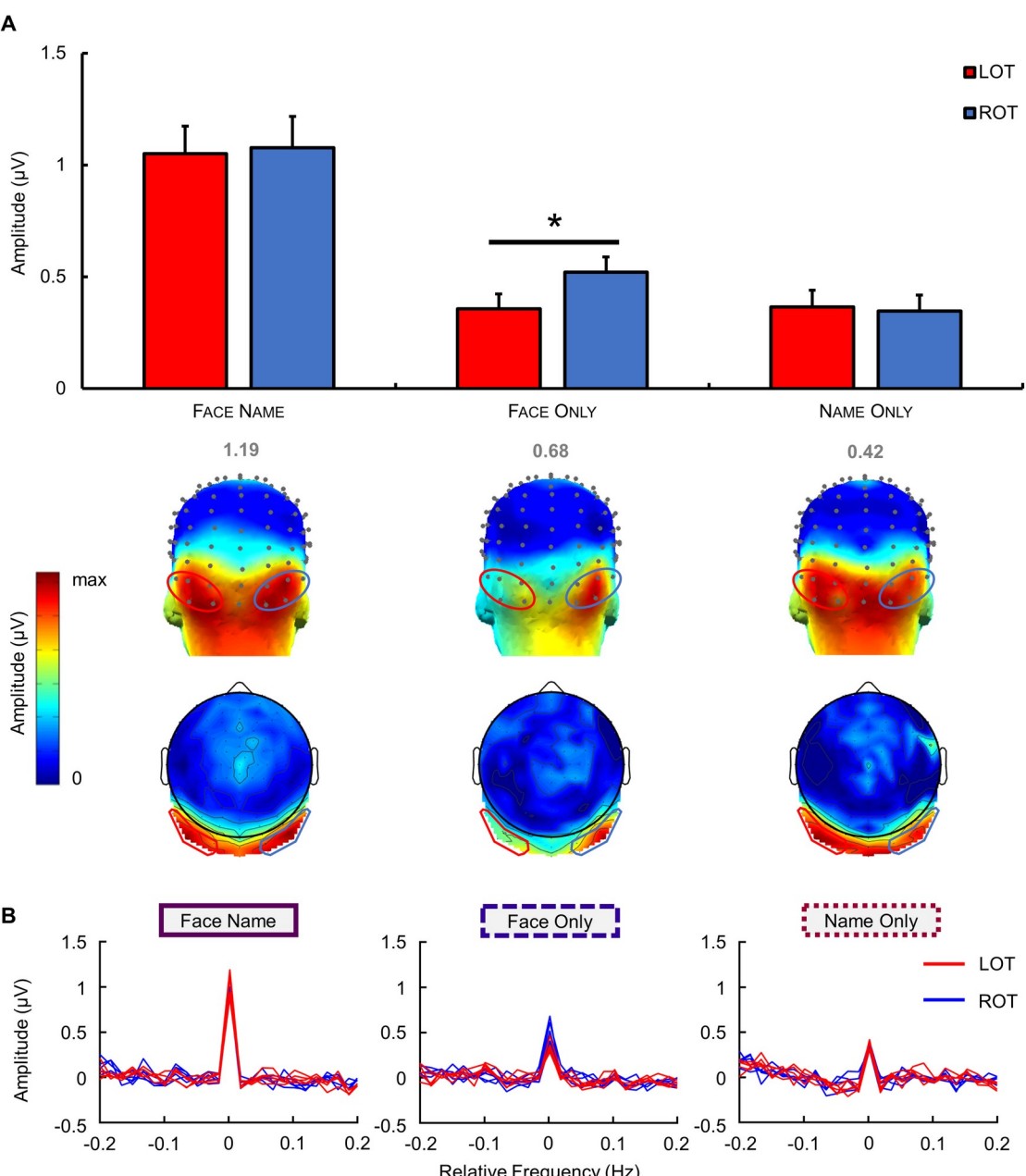

**Fig 4. Quantification and scalp topography of EEG responses in each condition of Experiment 2.** (A) Mean group-level identity-oddball responses in baseline-corrected amplitude (μV) at the two bilateral ROIs (shown with red and blue ovals on the topographical head to indicate the left and right ROI, respectively) regardless of identity. Error bars indicate standard error from the mean, reflecting variability across participants. Below, 3D and 2D topography maps show the distribution of oddball-identity responses at the sum of oddball harmonics for each condition at the group level (grand-averaged data, $n$ = 20 participants). Color scales' maxima are shown above each map, corresponding to the maximal baseline-corrected amplitude in μV. Asterisks indicate significant differences with a $p$-value $< 0.05$. (B) Chunked baseline-corrected amplitude spectrum for each condition at the sum of the six oddball harmonics at the group level. The middle of the chunked segment represents the oddball response over the sum of harmonics. LOT channels are represented in red; ROT channels are in blue. Data underlying this figure are deposited on a Dryad repository: https://doi.org/10.5061/dryad.m8t391m. EEG, electroencephalography; LOT, left occipito-temporal; ROI, region of interest; ROT, right occipito-temporal.

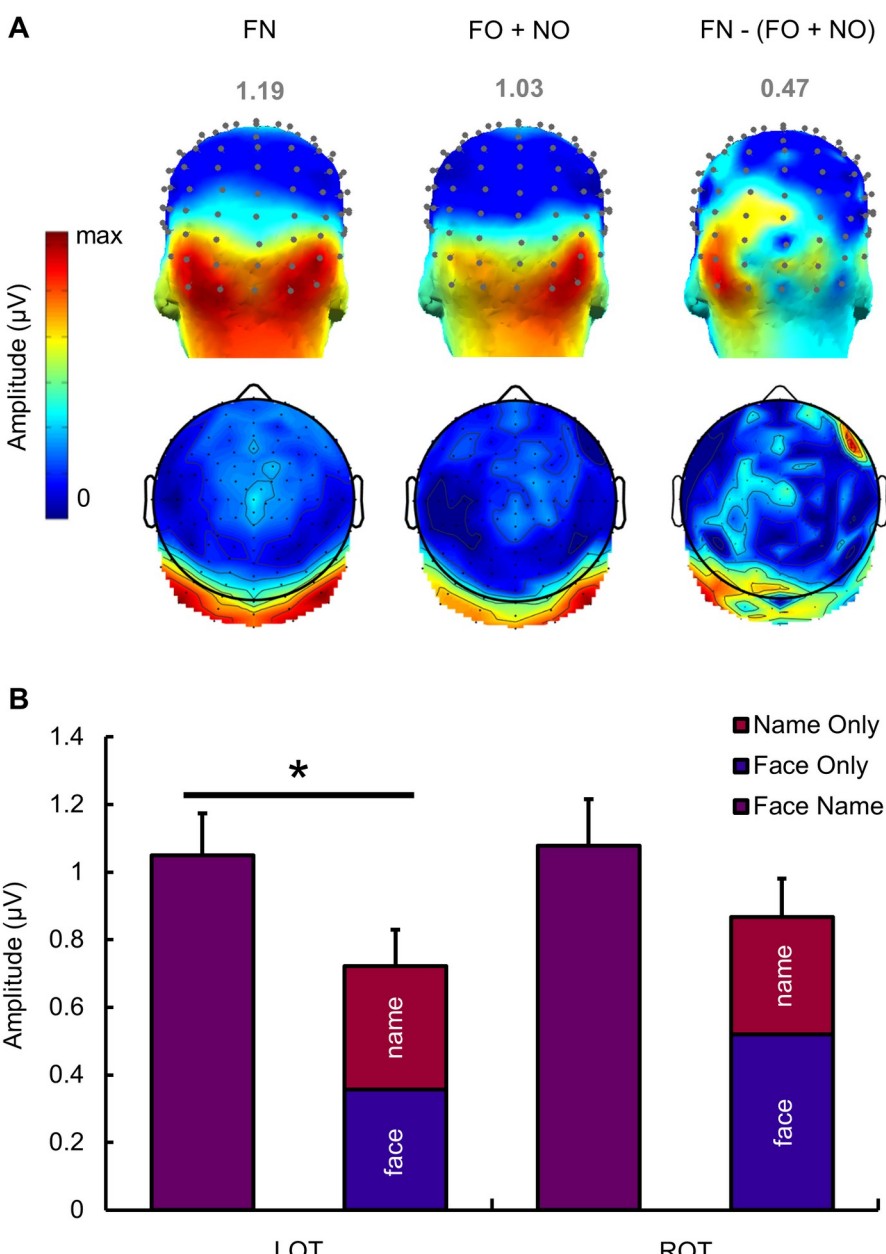

**Fig 5. Comparison of responses between the 3 conditions of Experiment 2 in scalp EEG.** (A) Three-dimensional (3D) and 2D group-level scalp topographies at the sum of identity-oddball harmonics (baseline-corrected amplitudes in µV) for the Face Name condition, the sum of Face Only and Name Only, and the subtraction of the responses at the sum of control conditions from the response at the Face Name condition. Color scales' maxima are shown above each map and correspond to the maximal amplitude for each condition. Although there is a small area responding at the frequency of interest over the right frontal region in the [FN − (FO + NO)] comparison, it is important to mention that this response is driven by one electrode only (electrode F6), while other electrodes nearby do not show any significant response. (B) Mean baseline-corrected amplitudes in µV at the sum of identity-oddball harmonics for the Face Name condition and the sum of the two control conditions Face Only and Name Only. Error bars indicate standard error from the mean. The asterisk indicates a significant difference at $p < 0.05$. Data underlying this figure are deposited on a Dryad repository: https://doi.org/10.5061/dryad.m8t391m. EEG, electroencephalography; FN, Face Name; FO, Face Only; NO, Name Only.

difference between the amplitude of the Face Name condition and the amplitude of the sum (FO + NO) was significant in the LOT ($t = 2.726$, $p = 0.013$) but not in the ROT ($t = 1.509$, $p = 0.148$). It is interesting to note that the amplitude of the response for the sum of control conditions (FO + NO) represented only 68% of the amplitude of the *Face Name* response in the left ROI, while it represented a greater extent (80%) of the Face Name response in the right ROI. That is, thanks to the straightforward quantification of the EEG response afforded by the FPVS approach, we were able to determine that about a third (32%) of the total Face Name response in the left hemisphere and a fifth (20%) of the total Face Name response in the right hemisphere was not explained by the sum of modality-specific representations elicited by control conditions.

**Base frequency responses.**   As in Experiment 1, a general visual response was expected at 4 Hz. Consistently, clear base responses were found in each condition. In the Face Name condition, the first to fifth base harmonics were highly significant ($p$-values < 0.001). In the Face Only condition, base responses were clearly visible up to the ninth base harmonic (9F = 35.991 Hz; $p$-values < 0.001). In the Name Only condition, the first to eighth base harmonics were statistically significant ($p$-values < 0.001, except for the sixth base harmonic: $p < 0.01$).

To quantify base EEG responses, we compared the baseline-corrected amplitudes (in μV) in the three main conditions at the sum of 9 base harmonics across all electrodes. A significant difference was found between conditions (F(1.386,26.340) = 8.561, $p = 0.004$). Post hoc tests showed a larger amplitude of the base response in the Face Only condition (1.39 μV ± 0.42) relative to the two other conditions (Face Name: 1.23 μV ± 0.42; Name Only: 1.26 μV ± 0.36; respectively, $p = 0.003$ and $p = 0.0002$).

We also contrasted baseline-corrected amplitudes (in μV) in each ROI for base responses in the 3 conditions. We found a significant effect of ROIs (F(1,19) = 8.498, $p = 0.009$, $d = 1.34$), showing a predominance of the response in the right hemisphere. There was also a significant effect of condition (F(2,38) = 14.325, $p < 0.001$, $d = 1.74$). Post hoc tests showed that the amplitude at the base frequency in the Face Only condition was greater than in the two other conditions (all $p$-values < 0.001), but there was no difference between the amplitude at the base frequency in the Face Name and the Name Only conditions ($p = 1$). There was no interaction effect between conditions and ROIs (F(2,38) = 0.883, $p = 0.422$, $d = 0.43$). These results suggest that the response at the base frequency was right lateralized, in line with the results of Experiment 1. Accordingly, the topographical scalp distribution of base frequency responses in Experiment 2 showed an overall distribution over the right middle occipital region (S4 Fig).

## Experiment 2b (intracerebral EEG)

Seven patients with refractory partial epilepsy were implanted with intracranial electrodes for clinical purposes and were tested with Experiment 2 (three conditions). Across 196 contacts implanted in the gray matter of the left ventral ATL (i.e., located anteriorly to the posterior tip of the hippocampus), we found 7 contacts showing significant responses at the frequency of person identity change (0.571 Hz) in at least one condition in 4 participants. The responses recorded on these 7 contacts and their anatomical locations in the individual anatomy are shown in S5 Fig (see also S6 Fig for their location in the Talairach space). One contact was found significant in the Name Only condition but not in the other Face Name and Face Only conditions (Talairach coordinates: $x = -66$, $y = -14$, $z = -19$; TM'11 in participant 5, located in the anterior part of the inferior and middle temporal gyri [antMTG/ITG]). One contact was found significant in both the Face Name and the Name Only conditions ($x = -39$, $y = -18$, $z = -16$; TM'4 in participant 5, located in the anterior fusiform gyrus [antFG]). Importantly, a

total of 5 contacts were found to elicit a significant response in the Face Name condition but no response in the two control conditions. However, for two of these contacts, the response was not significantly larger in the Face Name condition than when computing the sum of the responses in the other conditions [FN–(FO + NO)] (z < 1.65). These two contacts were respectively located in the antFG ($x = -42$, $y = -17$, $z = -16$; TM'5 in participant 5) and anterior segment of the occipito-temporal sulcus (antOTS; $x = -30$, $y = -15$, $z = -27$; TM'2 in participant 6).

Strikingly, three of these 5 contacts showed a "pure" face-name association response, i.e., a significant response in Face Name, no significant response in Face Only and Name Only, and a significant difference between Face Name and the sum (FO + NO). The responses recorded on these three contacts and their anatomical locations are shown in Fig 6 (see S6 Fig for their location in the Talairach space). They were located in a restricted anatomical region in the antFG ($x = -37$, $y = -7$, $z = -27$; TM'1 in participant 2) or its adjacent sulci (anterior segment of the collateral sulcus [antCoS], $x = -33$, $y = -33$, $z = -23$; TB'4 in participant 1; antOTS, $x = -34$, $y = -15$, $z = -25$; TM'3 in participant 6). It is important to note that these results were independent of the number of sequences viewed by the participants. Among the three participants showing pure responses in the Face Name condition, only one (participant 1) was tested twice (total of 12 sequences), while the two other participants were tested only once with the experiment (total of 6 sequences).

In order to assess whether pure face-name responses were specific to the ATL, we also considered responses on the contacts implanted posteriorly to the ATL, in the left posterior temporal lobe (PTL) of the same 7 participants (see S7 Fig for the topographic parcellation of the ventral occipito-temporal cortex [VOTC]). Across the 41 contacts implanted in the gray matter of this region, we found 8 contacts showing significant responses at the identity-oddball frequency in at least one condition in 3 participants, all located in the left fusiform gyrus. The responses recorded on these 8 contacts and their anatomical locations in the individual anatomy are shown in S8 Fig (see also S6 Fig for their location in the Talairach space). Importantly, no pure face-name response was found on any of these contacts. Two contacts were found to be significant in the Face Name condition and not in the Face Only and Name Only conditions, but the difference [FN–(FO + NO)] was not significant ($x = -31$ to $-35$, $y = -45$, $z = -19$; F'2 and F'3 in participant 2). One contact was significant in the Name Only condition but not in other conditions ($x = -34$, $y = -33$, $z = -21$; F'1 in participant 3). Three contiguous contacts in the same participant were significant in the Face Name and Face Only conditions but not in the Name Only condition ($x = -32$ to $-39$, $y = -41$, $z = -19$; F'3 to F'5 in participant 4). Two contacts were found to be significant in all 3 conditions ($x = -25$ to $-29$, $y = -41$, $z = -19$; F'1 and F'2 in participant 4).

Finally, to test whether there were local variations of the noise level that may explain why significant responses were found on some contacts and not on others, for each contact, we compared the standard deviation of the noise between all significant contacts across ATL and PTL ($N = 15$) and their adjacent nonsignificant contact ($N = 12$). There were no statistical differences regardless of the condition ($p$-value range: 0.84 to 0.96) (S1 Fig).

## Discussion

Using an original frequency-tagging approach with mixed modalities of stimulation while recording scalp EEG and intracerebral EEG (SEEG), we found a neural response at the specific frequency at which different famous identities, either as a written name or a face picture, interrupted the successive presentation of a repeated specific identity, also presented in either format. The response was identified objectively, i.e., exactly at the predicted frequency of oddball

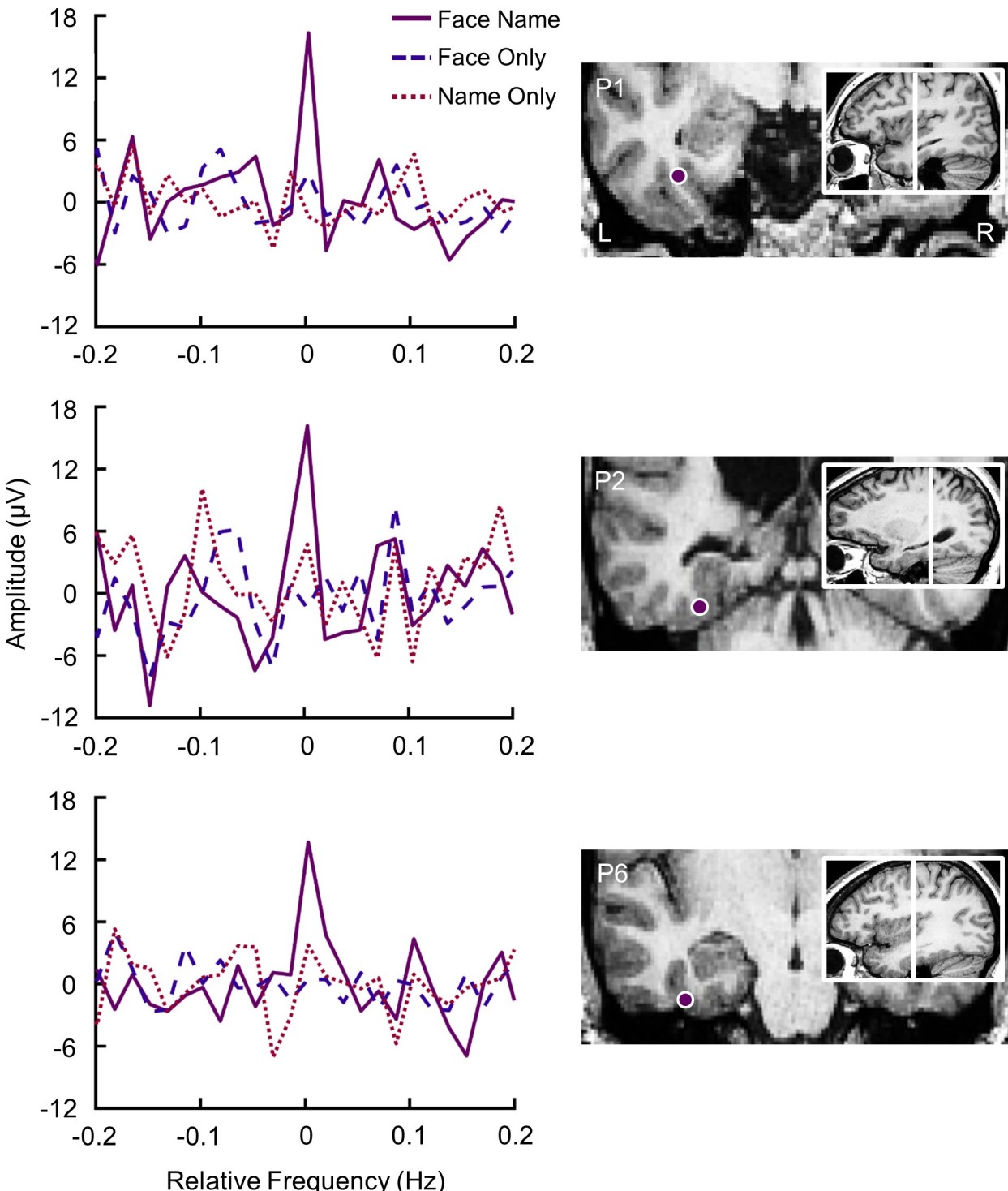

**Fig 6. Pure face-name (person identity) intracerebral responses in the ATL.** On the left, baseline-subtracted chunked FFT segments depicting oddball responses in Face Name, Face Only, and Name Only conditions as recorded at an individual contact. These contacts show a pure Face Name effect because (1) they show a strong electrophysiological response at the oddball frequency in the Face Name condition and (2) the sum of their responses in the two other conditions is significantly different from the response in the Face Name condition. On the right is shown the location of the individual contact responding selectively to the Face Name condition using a postimplantation CT scan co-registered to a preimplantation MRI. Data underlying this figure are deposited on a Dryad repository: https://doi.org/10.5061/dryad.m8t391m. ATL, anterior temporal lobe; FFT, fast Fourier transform.

presentation and its harmonics, over the bilateral occipito-temporal cortex in scalp EEG in Experiment 1. Thanks to another advantage of the fast periodic stimulation approach, i.e., its high signal-to-noise ratio, this response was clearly identified in 100% of the participants for the sum of harmonics across all electrodes of the bilateral ROIs and across the 4 famous identities tested.

To ensure that this crossmodal face-name response was not merely due to the linear superimposition of identity-oddball responses to faces and names generated by separated brain regions containing modality-specific person-related representations, we compared this response to the identity-oddball response elicited in 2 control conditions (Experiment 2, with 2 famous identities and 20 participants). In the Face Only condition, names of the base identity were replaced by other famous names, in order to isolate unimodal face responses that potentially contribute to the neural response observed in the Face Name condition. In the Name Only condition, face images of the base identity were replaced by other famous faces, isolating unimodal name responses potentially contributing to the neural face-name response. Critically, we found a Face Name response in the EEG spectrum significantly above and beyond the sum of the Face Only and Name Only responses [FN > (FO + NO)]. Moreover, thanks to frequency-tagging, we were able to quantify the contribution of the integrated response, which reached up to a third of the total EEG oddball response in the Face Name condition over the left hemisphere. Overall, these observations provide direct evidence for an integrated representation of face and name inputs of specific identities in the human brain. Finally, through intracerebral recordings in the left ventral ATL, we were able to identify a few electrode contacts with a unique response profile, i.e., showing a significant response in the Face Name condition only, suggesting that they capture neural activity associated with a multimodal integrated representation of person identity.

## Integrative hub or re-entrant interhemispheric connections?

Our hypotheses on the neural mechanism subtending face-name associations were grounded on two theories of the organization of semantic (here, person-related) knowledge: one that postulates separate modality-specific regions in each hemisphere [26,27], and the other postulating the existence of a common crossmodal region (integrative hub) in addition to modality-specific regions [43–45]. Although a large extent of the oddball EEG response recorded on the scalp (68% and 80% over the left and right hemispheres, respectively) point to modality-specific responses selective to a person identity, our observation of a significant and substantial nonlinearity (i.e., Face Name > Face Only + Name Only) clearly provides support for an integrative hub of face and name identity.

The exact neural sources of this effect, which was found over low occipito-temporal channels, are unknown, and may concern either or both of the PTLs and ATLs. Proponents of the unitary hub theory postulate that both hemispheres contribute to and are activated during multimodal semantic processing [44,45,52,53,55]. While the hub-and-spoke model has been revisited in favor of a graded functionalization of semantic representations according to the functional connectivity of the ATL subregions with other cerebral regions (e.g., more activation of the left hemisphere when language processes are involved) [49,50,73], our protocol did not require an explicit language-related task and used an equal number of verbal (written names) and pictorial (face pictures) stimuli. Hence, according to this view, whether the neural response recorded in scalp EEG originates from the ATL or more posterior regions, it should have engaged both hemispheres in a similar way. Interestingly, Face Name responses were significantly greater than the sum of Face Only and Name Only responses over left occipito-temporal electrodes in scalp EEG. However, there was also a trend for an effect in the same region

of the right hemisphere, and no significant interaction between conditions and hemispheres, so that overall, the EEG results cannot be taken as evidence against the view that both hemispheres equally hold an integrated ("hub") representation of faces and names. A more comprehensive recording of intracerebral responses across the bilateral occipito-temporal cortices with a paradigm as used here should shed more light on this issue of hemispheric lateralization in future investigations.

### The role of the (left) ATL

Our SEEG results are consistent with neuroimaging studies that have shown a common locus for famous face and name recognition in the ATL, even though face-name identity association per se was not investigated [28,45,59,60]. Moreover, several studies have found a specific role of the left temporal pole (anterior to the ATL in our study) in retrieving proper names when presented with a famous face or voice [74–76] and in providing semantic information when presented with a famous name [77]. However, these studies provided no direct evidence of an integration of person-related information at the level of identity. Here with SEEG, we recorded few but significant "pure" face-name responses in the left ventral ATL, i.e., not only a significant difference between Face Name and the sum (FO + NO) but a significant response in Face Name with no significant response in Face Only and Name Only.

Admittedly, the likelihood to find 3 pure face-name association responses among 196 ATL contacts sampled may appear to be very low. However, (1) these 3 pure face-name association responses were found in 3 different participants (i.e., in 3 out of 7 participants, in 43% of the participants); (2) the number of significant contacts depends on the size of the functional area (the smaller an area, the less likely it is to be sampled with electrodes and thus to record a significant response); and (3) the number of recorded contacts included in the proportion, and therefore the size of the ROI, is set arbitrary (our ATL region was wide, including lateral regions, i.e., the inferior and middle temporal gyri, and extending posteriorly until the posterior tip of the hippocampus, see S7 Fig). For example, restricting our ROI to the left antFG and adjacent sulci (medial bank of the antOTS and lateral bank of the antCOS), a region that is sampled by 1 or 2 SEEG electrodes per subject, leads to a proportion increase to 11.5% (3/26).

These pure face-name responses show that the left ventral ATL contains representations that are both crossmodal and sensitive to person identity. Presumably, in the Face Name condition, the neural activity must adapt or habituate to both faces and names during the presentation of a repeated identity and show a release from adaptation when a different identity is periodically presented, leading to a face-name oddball response. In the control conditions, the neural activity also adapts or habituates to the repeating faces or names of the specific identity but this adaptation is randomly (and not periodically) interrupted at every change of identity (at every oddball stimuli and at every other face and name replacing the specific identity in the sequence), leading to the absence of oddball response in these conditions. These results show that the left ventral ATL is a neural locus for the integration of a specific face and a name identity into a shared crossmodal representation. Thus, in concordance with studies showing a convergence of sensory information in the ATL [44,78], our results suggest that this ventral anterior temporal region receives visual information from both unimodal face and name regions. Moreover, our findings highlighting the integration of different input stimuli (face pictures and written names) into a semantic concept/identity in the ventral ATL are consistent with previous studies showing the importance of this region in semantic tasks irrespective of the input modality or stimulus category [55,73,79]. Nonetheless, it is important to point out that our findings shedding light on the left ventral ATL do not preclude a similar involvement of its right hemispheric counterpart, which was not explored in SEEG in the present study. We

therefore emphasize again the need for additional intracerebral recordings in larger samples of participants [80,81] in the bilateral ventral ATL in order to clarify its role in crossmodal person-related processing. In addition, recording of electrical activity of individual neurons [82] in the human ATL may provide more direct evidence for integration of identity-related faces and names at the neuronal level and provide useful contrasting information with previously recorded responses in the MTL [24].

### The relationship between scalp and intracerebral responses

We found a striking difference between responses recorded on the scalp and intracerebrally with SEEG. In scalp EEG, we found a significant response in the 3 conditions with up to a 32% increase in the amplitude elicited by the Face Name condition compared to the amplitude of the sum of control conditions over the left occipito-temporal cortex (and about 20% over the right homologous region). This shows that the bulk of the response (68% and 80% in the left and right hemispheres, respectively) still originates from separate modalities, suggesting the existence of modality-specific representations of identity in addition to a crossmodal representation. In SEEG, we recorded 5 contacts with a significant response in Face Name only and no significant response in Face Only and Name Only, including 3 contacts with significantly larger responses in Face Name compared to the sum of the two control conditions in the left ventral ATL. One plausible explanation for this discrepancy between the two approaches (recording of pure integrated responses inside the brain but not on the scalp) is that scalp EEG recorded a combination of responses coming from modality-specific regions and pure face-name association responses coming from crossmodal regions coding shared person-related representations.

### Clinical relevance of the FPVS approach

Semantic impairments are common in the neurological population, and face-name associations are particularly susceptible to be impaired. As mentioned in the introduction, several neurological conditions are frequently associated with face-name association deficits (e.g., SD, Alzheimer disease, etc.). However, face-name association deficits are difficult to evaluate in these neurological populations since the tests that are commonly used (e.g., the Iowa Famous Faces Test [83–85] or the Boston Famous Faces test [86,87]) require naming and decisional processes in addition to semantic processing. Moreover, these tests require explicit tasks that can be misunderstood by participants who may have comprehension difficulties (e.g., due to a low IQ). Our FPVS-EEG technique provides an objective measure of face-name association processes that do not require to name faces or to ask for an explicit task or an explicit semantic decision or response. FPVS-EEG could therefore be a method of choice to diagnose face-name association deficits in neurological populations, for example, in patients with SD characterized by a degradation of crossmodal anterior temporal regions, in whom the nonlinear pattern found in typical brains (FN > FO + NO) could be reduced or absent, with more linear patterns of responses (FN = FO + NO) observed.

### Materials and methods

For clarity purposes, this section is divided in three parts. First, we describe the materials and methods for Experiment 1 (Face Name condition alone, 4 facial identities) administered to normal participants in scalp EEG. In the second and third sections, we describe materials and methods for Experiment 2 (Face Name condition and two control conditions, 2 facial identities) applied to neurotypical participants in scalp EEG and to epileptic participants in intracerebral EEG (SEEG), respectively.

### Ethics statement

All subjects gave written informed consent to participate, and the study was approved by the Biomedical Ethical Committee of the University of Louvain (B403201111965) (scalp EEG) and by the Research Ethics Committee of the CHRU-Nancy (2015-A01951-48) (SEEG). The experiments were performed in compliance with the Declaration of Helsinki.

### Experiment 1

**Participants.** Twelve volunteers (6 females, all right-handed, mean age = 22.61 years, standard deviation = 2.13 years) were tested individually and received financial compensation in exchange for their participation. All participants reported normal or corrected-to-normal vision.

**Stimuli.** Four highly famous male identities, two politicians (Donald Trump and Vladimir Putin) and two actors (Brad Pitt and Johnny Depp), were selected. For each of them, 18 natural photographs (depicting different head orientations, lighting conditions, facial expressions, and background) were selected from the internet. The stimuli were selected to be as variable as possible to minimize repetition effects due to low-level features [88]. For each celebrity, the written names were presented on a gray background in capital letters but were made variable by using 18 different fonts and six different colors. Thirty-six other famous identities were also selected (18 pictures and 18 written names) to create one stimulus per identity (either a face or a name). Names of these other celebrities were also written in 18 fonts, different from those used with the four main identities, and the same six colors were chosen (red, black, green, brown, yellow, and blue). Face pictures were resized to 200 × 250 pixels.

**FPVS procedure.** Brain activity of each participant was recorded with high-density (128 channels) scalp EEG while participants were seating in front of a computer screen at a distance of 87 cm. To help participants pay attention to the screen before each sequence, a fixation cross was first displayed on the uniform gray background for 2–5 seconds (this duration randomly varied between sequences). A 60-second sequence was then run, plus 2 seconds of fading time before and after the sequence to avoid potential blinks and eye movements due to the sudden appearance or disappearance of flickering stimuli. During the fade-in, the contrast modulation depth of the periodic stimulation progressively increased from 0% to 100% to reach full contrast while it decreased during the fade-out. Across sequences, participants were presented with stimuli through sinusoidal contrast modulation from 0% to 100% at a rate of 4 Hz (= 4 stimuli per second). A relatively low stimulation frequency of 4 Hz (compared to higher stimulation frequencies at 6 Hz used in face categorization studies [88]) was selected based on pilot data and to provide sufficient stimulus duration to extract information necessary to associate specific face and name identities. Within a sequence, each stimulation cycle lasted 250 ms (i.e., 1,000 ms/4) in which an image appeared and disappeared in the middle of the screen as its contrast followed a sinusoidal function (Fig 1). Image size varied randomly at each stimulus presentation cycle across five equidistant steps between 80% and 120%, in order to further reduce the potential impact of low-level adaptation. At 100%, face images subtended about 6.78˚ × 5.47˚ of visual angle, while words subtended on average 13.03˚ × 1.41˚ of visual angle.

The stimulation was repeated once for each of the four famous target identity, representing a total of eight sequences. Each sequence was composed of base and identity-oddball stimuli. Base stimuli were highly variable photographs and written names corresponding to one of the four famous identities (B. Pitt, J. Depp, D. Trump, and V. Putin), while identity-oddball stimuli were variable photographs or names of different identities (Fig 1). Within sequences, all images were randomly selected from the pool of images of their respective types (base stimuli

or identity-oddball stimuli). All images appeared at a frequency of 4 Hz (base frequency, reflecting a general visual processing) and identity-oddball stimuli were displayed every seven stimuli, i.e., every 0.571 Hz (4 Hz / 7; identity-oddball frequency, thought to reflect the disruption from the flow of same-identity stimuli).

Four blocks of eight stimulation sequences were defined prior to the beginning of the experiment, each with a fixed, counterbalanced order of sequences within a block, so that randomization uncertainties due to a small number of participants were avoided. Blocks were administered one after another, every subject being presented with only one block (subject 1 was presented with block 1, subject 2 with block 2, . . . subject 5 with block 1, etc.). Within a block, participants were invited to rest their eyes between every trial.

Participants were aware that they would be presented with famous faces and names, but no further information was given about the procedure. They were asked not to pay particular attention to these stimuli and rather look at the blue fixation cross located in the middle of the screen. Their task was to press the space bar of the keyboard when the fixation cross turned to red. Color changes randomly occurred 8 times for 500 ms within every sequence. This task was orthogonal to the manipulation of interest and was used to ensure that participants maintain a constant level of attention throughout the experiment. Mean accuracy and response time (RT) to the orthogonal task (detecting color changes of the fixation cross) were computed independently for each participant. Behavioral performance was close to ceiling, with a mean accuracy of 98.31% (standard deviation = 2.05%) and mean RTs around 502 ms (standard deviation = 59 ms).

At the end of the experiment, participants were provided with a questionnaire depicting 44 black-and-white faces (the 4 famous base identities, the 36 famous identity-oddball identities, and 4 unknown distractors). This questionnaire was designed to ensure that target identities were well known to the participants. They had to tell if they knew the depicted celebrity (yes, no, uncertain) and for each one, to rate its familiarity on a 5-point scale (ranging from rarely seen to very frequently seen on TV or other media). They also had to tell if they had noticed this face during the experiment or not. Overall, 89.8% (standard deviation = 15%) of the 40 famous faces (4 base identities and 36 identity-oddball stimuli) were rated as known by the participants. Considering base identities only, participants had no difficulty recognizing them, with perfect recognition for D. Trump, J. Depp, and V. Putin (recognized for 100% of the participants), and near perfect recognition for B. Pitt (recognized by 91.6% of the participants; 11/12). None of the four unknown faces (not used in the experiment) were rated as known. When rating face familiarity, participants indicated that they were frequently exposed to these famous people in the media (mean rating of 3.74 on a 5-point scale, with 1 = rarely, 5 = many times; standard deviation = 0.61). D. Trump, J. Depp, and V. Putin were reported as seen by all the participants during testing. Only one participant did not report having seen Brad Pitt during the course of the experiment and this was related to the nonrecognition of this famous face as represented in the black-and-white questionnaire.

**EEG recording.**   EEG was recorded using a 128-channels Biosemi ActiveTwo system. The system uses two additional electrodes for reference (CMS, common mode sense) and ground (DRL, driven right leg). Vertical and horizontal electrooculogram (EOG) was recorded by placing two flat-type electrodes above and below the participant's right eye and two at the outer canthi of the eyes. The EEG and EOG were digitized at a sampling rate of 512 Hz. Recordings were manually initiated when participants showed an artifact-free EEG signal. To facilitate analyses, Biosemi channel labels were later converted to an extended 10–20 system of electrode placement [69].

**EEG analysis.**   Preprocessing of EEG data was carried out using Letswave 5 (https://github.com/NOCIONS/Letswave5), a custom software running over Matlab R2012b

(MathWorks, Natick, MA), and followed procedure developed for frequency tagging studies of face stimuli in a number of published studies [68,71,89]. After importation of raw EEG data files, vertical jumps in the signal due to voltage drift during pauses between trials were corrected for by aligning continuous blocks of recording to the first block. EEG data were band-pass filtered (0.1 to 100-Hz fourth-order zero-phase Butterworth filter) and multi-notch filtered (width 0.5 Hz) to remove 50 Hz's electrical noise on four harmonics. A downsampling to 256 Hz was applied to reduce file size and increase processing speed. The data were then segmented using four distinct stimulation start triggers (one per famous base identity) into epochs from 2 seconds before stimulation to 2 seconds after the end of stimulation (−2 seconds to 66 seconds). A blink detection was applied and a threshold of more than 0.15 blinks per second for each epoch was used in order to apply correction of eye movements when needed [71]. Following this threshold, an ICA (Independent Component Analysis) was computed for three subjects. When needed, noisy electrodes across multiple trials (less than 5% of channels) were linearly interpolated with 3 to 6 pooled neighboring channels. Epochs were then re-referenced to the common average.

Frequency domain analyses were carried out by further segmenting epochs using an integer number of bins of 0.571 Hz cycles beginning 2 seconds after onset of the stimulation sequence (then removing fade-in artifact-prone responses) until approximately 60 seconds, before stimulus fade-out (15.235 bins in total). Resulting epochs were then averaged to improve signal-to-noise ratio. A fast Fourier transform (FFT) was computed and amplitude spectra were extracted. Data were grand-averaged for each channel across participants.

A baseline-subtraction was computed to account for differences in baseline noise across the frequency spectrum and to quantify the electrophysiological responses in microvolts [90]. The difference between the amplitude of the bin of interest and the average of amplitude in the 20 surrounding bins was computed, excluding the immediately adjacent bin in case of remaining spectral leakage, and the local maximum and minimum amplitude bins to avoid projecting the signal in the neighboring bins containing noise. To quantify the responses at the frequency of identity change, we computed the sum of identity-oddball harmonics (0.571 Hz, 1.142, and so on until the sixth harmonic, i.e., 3.427 Hz). To do so, responses on the FFT spectrum were segmented into 6 separate chunks centered on the bin containing the harmonic of interest (see also S9B Fig illustrating the same procedure with SEEG data). Each chunk contained 31 bins (i.e., a chunk length of 0.5208 Hz), with the bin in the middle (the 16th bin, with 15 bins on each side) corresponding to the identity-oddball frequency. The 6 chunks were then summed for each participant (see also S9C Fig), and a grand-average was computed across participants. The quantification of the response at the sum of harmonics was computed following the same principle as for the quantification of responses on the whole FFT spectrum, i.e., with a baseline-subtraction.

In order to determine whether a significant response was present at the frequencies of interest and harmonics, Z-scores were calculated by computing the difference between the amplitude at the bin of interest and the mean amplitude in the 22 surrounding bins (11 bins on either side, no exclusion of the local maximum and minimum amplitude bins) and dividing this value by the standard deviation of amplitudes in the 22 corresponding surrounding bins [68]. An electrode was considered as showing a significant response if the Z-score at the frequency bin of the identity-oddball stimulation exceeded 1.65 (i.e., $p < 0.05$).

Following the analysis of amplitudes, we examined the overall scalp topography at base and identity-oddball frequencies.

### Experiment 2a (scalp EEG)

Experiment 1 was designed as a proof of concept and to identify facial identities providing large responses in the paradigm. Experiment 2 used only two facial identities (D. Trump and

V. Putin) and increased variability in written stimuli to counter low-level effects. Most importantly, it added two key control conditions to identify the role of modality-specific representations within the face-name sequence.

**Participants.** Twenty-two volunteers (10 males, all right-handed) received financial compensation in exchange for their participation in the experiment. One male and one female participant were excluded from the EEG analyses because of excessive artifacts in a large set of electrodes across multiple epochs during data recording. The final sampling of participants ($n$ = 20) had a mean age of 22.96 years (standard deviation = 1.91 years). All participants reported normal or corrected-to-normal vision.

**Stimuli.** Images and names of two famous identities (D. Trump and V. Putin) were used as base identities. The same 18 natural photographs as in Experiment 1 were used, but the variability among written names was increased by using both uppercase and lowercase letters, and less conventional fonts (e.g., Ravie, Pristina, Segoe Script, etc., instead of Arial, Candara, etc.). A gray background and the same six colors as in Experiment 1 were used for the written stimuli. Seventy-two other famous identities were also selected, 36 being the same as in Experiment 1 so that oddball stimuli were unchanged, and 36 novel ones, serving as base control stimuli. Thirty-six face photographs were used, each depicting a single famous face, with different viewpoints, lighting conditions, background, and facial expressions (18 identity-oddball faces and 18 base control faces). Face pictures were resized to 200 × 250 pixels. Thirty-six written stimuli were created, with 18 using the same design criteria as those of the base identities (= control names) and 18 being written in different fonts (= identity-oddball names).

**FPVS procedure.** The paradigm and testing conditions were similar to Experiment 1, with the two control conditions added. The experiment consisted of three conditions for each famous base identity, i.e., a total of 6 different sequences (3 conditions × 2 identities).

In the Face Name condition, base and identity-oddball stimuli were similar to those used in Experiment 1, with the exception that only D. Trump and V. Putin appeared as base stimuli and that written stimuli were even more variable in terms of size, color and font (Fig 1). In the Face Only condition, famous base faces were unchanged, but famous base names were replaced by 18 different famous names. Identity-oddball stimuli were the same than in the Face Name condition (Fig 1). This condition aimed at controlling the amount of modality (face)-specific representations that are activated in the face-name association sequence. If a response is observed at the identity-oddball frequency in this condition (0.571 Hz), it cannot reflect the disruption of face-name associative representations (i.e., D. Trump's name with his face) but can reflect the fact that 50% (on average) of the stimuli appearing every 7 stimuli are faces of a different identity than the base face identity (D. Trump). The reverse was done in the Name Only condition in which only famous base names were unchanged, while famous base faces were replaced by 18 different famous faces. Identity-oddball stimuli remained unchanged according to the Face Name condition (Fig 1). The purpose of this condition was to quantify and isolate the amount of modality(name)-specific representations that are elicited by the face-name paradigm. If a response is observed in this condition at the identity-oddball frequency, it can reflect a disruption in the activation of name representations due to the occurrence of different identity names among identity-oddball stimuli.

Each sequence was presented once within a block, and the order of sequences was counterbalanced. Each subject was presented with two blocks, i.e., two presentations per sequence. Within sequences, all images were randomly selected from the pool of images of their respective types (base stimuli or identity-oddball stimuli). All images appeared at a frequency of 4 Hz (base frequency, reflecting a general visual processing) and identity-oddball stimuli appeared every six base stimuli, i.e., every 0.571 Hz. At 100% of their size, face images subtended about 6.78˚ × 5.47˚ of visual angle, while words subtended on average 13.65˚ × 1.62˚ of visual angle.

Mean accuracy and RT to the orthogonal task (detecting color changes of the fixation cross) were computed independently for each participant. We considered behavioral data across identities in the three main conditions: Face Name, Face Only, and Name Only. A one-way ANOVA with repeated measures found no difference between conditions according to accuracy ($F_{(2,38)} = 0.241$, $p = 0.787$) or RTs ($F_{(2,38)} = 0.148$, $p = 0.863$).

Participants also had to complete the same questionnaire as in Experiment 1. Here, 89.25% (standard deviation = 15.75%) of the 40 famous faces were rated as known. The two base identities (D. Trump and V. Putin) were rated as known by all the participants. Overall, some unknown faces were sometimes reported as known, with a 5% occurrence among participants. In these very few cases the supposedly unknown face was globally rated as less frequently seen in the media (mean = 2.83), but the results presented a large variability (standard deviation = 1.61). When asking participants more information about these faces, they reported incorrect information, suggesting confusion and misrecognition of the faces. When rating face familiarity, participants reported a frequent encounter of famous faces in the media (mean = 3.64; standard deviation = 0.53). All the participants reported having seen D. Trump, and only one participant did not rate V. Putin as seen during the testing.

**EEG recording and analysis.** EEG recordings and analyses were performed exactly as in Experiment 1, considering the Face Name, Face Only, and Name Only conditions, regardless of identity.

## Experiment 2b (intracerebral EEG)

**Participants.** Considering the selective left-lateralized face-name responses observed in scalp EEG and patients' availability, we selected the left ATL as ROI. We recruited 7 participants (6 females, mean age: 30.7 ± 12.5 years, 4 right-handed) undergoing clinical intracerebral evaluation with depth electrodes (SEEG [91]) for refractory partial epilepsy (Epilepsy Unit, University Hospital of Nancy). Participants with at least one intracerebral electrode implanted in the left ventral ATL were included. The ventral ATL was defined as the ventral region located anteriorly to the posterior tip of the hippocampus (see S7 Fig for the topographic parcellation of the VOTC used in this study) [80,81]. The ventral ATL comprises the following anatomical regions: the anterior segment of the parahippocampal gyrus (antPHG), the antCoS, the antFG, the antOTS, and the antMTG/ITG.

**Intracerebral EEG recording.** Intracerebral electrodes were stereotactically implanted within the participants' brains for clinical purposes, i.e., in order to delineate their seizure onset zones. SEEG avoids the need for large craniotomies as in the more popular implantation of subdural grid electrodes to record intracranial EEG (electrocorticography or ECoG) but offers the possibility to accurately explore mesial structures, deep sulci, and the insula [91]. Due to its intrinsic precision placement features, SEEG may be associated with fewer complications than implantation of subdural grid electrodes [92]. A recent meta-analysis provides strong data regarding the safety of SEEG based on a systematic review of all published complications [92–94]. The implantation scheme is defined individually for each patient according to electro-clinical hypotheses derived from noninvasive investigation; the number of implanted electrodes is minimized, and their location and trajectories are defined strictly based on clinical criteria. After induction of general anesthesia, the stereotactic frame (Leksell model G, Elekta instrument, Stockholm, Sweden) is positioned on the patient's head. A stereotactic CT scan is then performed and fused to the preoperative non-stereotactic MRI using the iPlan Stereotaxy software (Brainlab AG, München, Germany). Electrodes are then implanted according to the following procedure: After reporting the calculated coordinates on the frame, stereotactic guided drilling of the skull is performed and a bone screw is inserted. The

intracerebral electrode (DixiMedical, Besançon, France; diameter of 0.8 mm; 5 to 18 platinum/iridium contacts with 2-mm length, 1.5 mm apart) is inserted and secured to the screw with a tight seal in order to prevent cerebrospinal fluid leak. Fluoroscopic control is performed to rule out major electrode positioning errors before removing the stereotactic frame. A safety margin of 2 mm is respected between the trajectory and the nearby vessels [94].

A total of 196 contacts (mean number per participant = 28; range = 11–39) were distributed over the left ventral ATL of 7 participants, and 41 contacts (mean number per participant = 10; range = 10–11) were implanted over the left PTL in 4 participants. Intracerebral EEG was sampled at a 512 Hz and referenced to either a midline prefrontal scalp electrode (FPz, available in 3 participants) or an intracerebral contact in the white matter (in 4 participants).

**FPVS procedure.**   Stimuli and the FPVS procedure were the same as in Experiment 2 in scalp EEG, except the number of blocks. All participants were administered with at least 6 sequences (2 Face Name, 2 Face Only, and 2 Name Only), corresponding to one session of the experiment. For 4 participants, the experiment was administered twice, corresponding to a total of 12 sequences (4 Face Name, 4 Face Only, and 4 Name Only).

Mean accuracy and RT to the fixation cross task were calculated for each participant. A one-way ANOVA with repeated measures found no difference between conditions according to accuracy ($F(2,16) = 1.019$, $p = 0.383$) or RTs ($F(2,16) = 0.326$, $p = 0.726$).

**Intracerebral EEG analysis.**   SEEG analyses were carried out using the free software Letswave 5 (see https://github.com/NOCIONS/letswave6 for the most recent version) and largely followed procedures used in previous SEEG studies with this approach [80,81]. Segments of SEEG data corresponding to stimulation sequences were extracted. These segments were then cropped to an integer number of 0.571 Hz cycles beginning after the 2-second fade-in and ending before the 2-second fade-out (30,471 bins, approximately 60 seconds). Sequences were averaged in the time domain separately for each condition and each participant. No further preprocessing was applied to the recordings. Subsequently, a FFT was applied to these averaged segments.

Responses significantly above noise level in the frequency domain at the identity-oddball stimulation frequency (0.571 Hz) and its harmonics were determined in each condition as follows: (1) the FFT spectrum was cut into segments of 0.8 Hz centered at the identity-oddball response frequency and the 6 first harmonics, i.e., 0.571 Hz, 1.142 Hz, 1.714 Hz, etc. (S9B Fig); (2) the amplitude values of these 6 FFT segments were summed (S9C Fig); and (3) the summed FFT spectrum was transformed into a Z-score (S9D Fig). Z-scores were computed as the difference between the amplitude at the identity-oddball frequency bin and the mean amplitude of 22 surrounding bins (12 bins on each side, excluding the first bin directly adjacent to the bin of interest) divided by the standard deviation of amplitudes in the corresponding 22 surrounding bins [80]. A contact was considered as showing a significant response in a given condition if the Z-score at the bin of the identity-oddball frequency exceeded 3.1 (i.e., $p < 0.001$ one-tailed: signal > noise; a conservative threshold is used to take into account the number of contacts and harmonics tested).

Identity-oddball amplitude responses were quantified using baseline-subtracted amplitudes. Baseline-corrected amplitudes were computed as the difference between the amplitude at each frequency bin and the average of 22 surrounding bins (12 bins on each side, excluding the first bin directly adjacent to the bin of interest). Amplitude responses were quantified as the sum of the baseline-subtracted amplitudes at the identity-oddball frequencies from the first until the sixth harmonic (0.571 Hz until 3.428 Hz), separately for each condition.

At each contact, the amplitude in the Face Name condition was compared to the sum of amplitudes in the Face Only and Name Only conditions. To do so, for each contact, we summed the 2 FFT spectra corresponding to the conditions Face Only and Name Only (S9E

Fig), and we subtracted this resulting summed FFT spectrum from the FFT spectrum corresponding to the Face Name condition (S9F Fig). Then, to assess the significance of this subtraction, we followed the same procedure as for single conditions: (1) the subtracted FFT spectrum was cut into segments of 0.8 Hz centered at the identity-oddball response frequency and the six first harmonics, i.e., 0.571 Hz, 1.142 Hz, 1.714 Hz, etc. (S9F Fig); (2) the amplitude values of these 6 FFT segments were summed (S9G Fig); and (3) the summed FFT spectrum was transformed into a Z-score (S9H Fig). Z-scores were computed as the difference between the amplitude at the identity-oddball frequency bin and the mean amplitude of 22 surrounding bins (12 bins on each side, excluding the first bin directly adjacent to the bin of interest) divided by the standard deviation of amplitudes in the corresponding 22 surrounding bins [80]. A contact was considered as showing a larger response for the Face Name than for the sum Face Only + Face Name if the Z-score at the frequency bin of the identity-oddball stimulation exceeded 1.65 (i.e., $p < 0.05$, one-tailed).

**Contact localization in the individual anatomy.** The exact position of each contact in the individual anatomy was determined for each participant by co-registration of the postoperative CT scan with a T1-weighted MRI of the participant's head. To accurately label each contact in the individual anatomy, we used the same topographic parcellation of the VOTC as in previous studies [81,95] (S7 Fig). Major VOTC sulci (collateral sulcus and occipito-temporal sulcus) served as mediolateral landmarks, while coronal reference planes containing given landmarks served as posteroanterior landmarks. A coronal plane including the anterior tip of the parieto-occipital sulcus served as the border of the occipital lobe and PTL. A coronal plane including the posterior tip of the hippocampus served as the border between the PTL and the ATL. Therefore, contacts located anteriorly to the posterior tip of the hippocampus were labeled in the ATL.

All EEG and SEEG data are deposited in a Dryad repository: https://doi.org/10.5061/dryad.m8t391m [96].

## Statistical testing

To compare effects between conditions or ROIs, one- or two-way ANOVA with repeated measures (Greenhouse-Geisser corrected when sphericity was violated) and paired *t* tests were computed when assumptions of normality were respected. Bonferroni-corrected post hoc tests were applied when ANOVA showed statistical differences between groups. When assumptions of normality were violated, nonparametric Friedman tests and Wilcoxon signed-rank tests (Bonferroni adjusted for multiple comparisons) were carried out. Effect sizes (Cohen's *d*) are provided.

## Supporting information

**S1 Fig. Noise comparison across conditions and regions in scalp EEG and SEEG.** In Experiment 1, the mean standard deviation of the noise around the identity-oddball frequency is displayed for the left, middle, and right ROIs (3 electrodes in each). In Experiment 2 with scalp EEG, the mean standard deviation of the noise in the left and right ROIs (5 electrodes each) is shown for the 3 conditions: Face Name, Face Only, and Name Only. In Experiment 2 with intracerebral EEG, the mean standard deviation of the noise around the identity-oddball bin is shown for all significant contacts in the ATL and PTL ($N = 15$) and for all their nonsignificant adjacent contacts ($N = 12$) across all the 6 participants showing a significant response in at least one condition, either in the ATL or PTL. Error bars indicate standard deviation of the mean. Data underlying this figure are deposited on a Dryad repository: https://doi.org/10.5061/dryad.m8t391m. ATL, anterior temporal lobe; EEG, electroencephalography; LOT, left

occipito-temporal; mid OT, middle occipito-temporal; PTL, posterior temporal lobe; ROI, region of interest; ROT, right occipito-temporal; SEEG, stereo electroencephalography. (TIF)

**S2 Fig. Scalp EEG responses at the individual level in Experiment 1.** Individual participant data for the 12 participants of Experiment 1 are shown at the frequency of person identity change (sum of 6 harmonics), independently of base identity. Each topography and the wave-form to its right correspond to the data of one participant. Head plots are scaled from 0 μV (dark blue) to the voltage of the maximal channel (red), separately for each participant. Wave-forms show the average of baseline-corrected chunked frequency spectrum of the 6 channels in bilateral occipito-temporal ROIs (P10, PO10, and PO12 for the LOT, and P9, PO9, and PO11 for the ROT), centered on the sum of harmonics, with an $x$ axis of relative frequency in Hz and a $y$ axis of amplitude (μV). All participants showed a significant response ($p < 0.01$) at the frequency of person identity change in the bilateral ROIs. Data underlying this figure are deposited on a Dryad repository: https://doi.org/10.5061/dryad.m8t391m. EEG, electroencephalography; LOT, left occipito-temporal; ROI, region of interest; ROT, right occipito-temporal. (TIF)

**S3 Fig. FFT spectrum of scalp EEG responses in Experiment 2.** Baseline-corrected FFT spectra (in μV) of the responses in the 3 conditions of Experiment 2 are shown at the group level ($n = 20$ participants). The red and blue lines represent the average of the 5 electrodes in the left and right ROIs, respectively; the location of the ROI is indicated by the topographical head on the top of each spectrum. Black labels on the FFT spectrum signal the significant oddball frequencies; the light gray label indicates the base frequency. Data underlying this figure are deposited on a Dryad repository: https://doi.org/10.5061/dryad.m8t391m. EEG, electroencephalography; FFT, fast Fourier transform; ROI, region of interest. (TIF)

**S4 Fig. Scalp EEG responses at the base frequency in Experiment 2.** Base frequency responses (sum of the first 9 base harmonics) are shown for each of the 3 conditions: Face Name, Face Only, and Name Only. Above, mean group-level base responses in baseline-corrected amplitude (μV) at the two lateral ROIs regardless of identity. Error bars indicate standard error from the mean, reflecting variability across participants. Asterisks indicate significant differences at p < 0.05. Below, topography of the response at the base frequency for each condition. Red and blue ovals show the electrodes included in the left and right ROIs, respectively. The color scale maximum is shown in light gray above each map and corresponds to the maximal baseline-corrected amplitude (in μV) in each condition. Data underlying this figure are deposited on a Dryad repository: https://doi.org/10.5061/dryad.m8t391m. EEG, electroencephalography; LOT, left occipito-temporal; ROI, region of interest; ROT, right occipito-temporal. (TIF)

**S5 Fig. Anatomical location and SEEG responses of the significant contacts in the ATL.** Anatomical location and electrophysiological responses of the 7 significant contacts in the ATL in 4 participants (P1, P2, P5, P6) and of their adjacent contacts. The significant contacts are highlighted in red. Significant base responses were determined in the same way as significant identity-oddball responses by (1) epoching the EEG frequency spectrum into segments centered on the first 6 base harmonics (i.e., 4 Hz, 8 Hz, etc.); (2) summing the amplitude values of these 6 frequency spectra segments; and (3) transforming it into a Z-score (difference between the amplitude at the base frequency bin and the mean amplitude of the 22

surrounding bins, divided by the standard deviation of amplitudes in the corresponding 22 surrounding bins). Data underlying this figure are deposited on a Dryad repository: https://doi.org/10.5061/dryad.m8t391m. ATL, anterior temporal lobe; EEG, electroencephalography; SEEG, stereo electroencephalography
(TIF)

**S6 Fig. Spatial distribution of significant and nonsignificant intracerebral contacts in the Talairach space.** Map of all 237 VOTC recording contacts implanted in the gray matter of the left ATL and PTL across the 7 participants displayed in the Talairach space using a transparent reconstructed cortical surface of the Colin27 brain (left hemisphere, ventral view). Each circle represents a single recording contact. Color-filled circles correspond to significant contacts in at least one condition. Significant contacts are color-coded according to the condition(s) for which we recorded significant responses at the oddball-identity frequency ($p < 0.001$). "Pure FN" contacts are contacts on which the response was significant in the Face Name condition but not in the Face Only and Name Only conditions, and the response in the Face Name condition was larger ($p < 0.05$) than the response in the sum of the two control conditions. "Non pure" FN contacts are contacts on which the response was significant in the Face Name condition but not in the Face Only and Name Only conditions, but the response in the Face Name condition was not significantly larger than the sum of the two control conditions. Note that the more posterior "Pure" FN contact is still located in the ATL (i.e., anterior to the posterior tip of the hippocampus in the individual anatomy). Data underlying this figure are deposited on a Dryad repository: https://doi.org/10.5061/dryad.m8t391m. ATL, anterior temporal lobe; FN, Face Name; FO, Face Only; NO, Name Only; PTL, posterior temporal lobe; VOTC, ventral occipito-temporal cortex.
(TIF)

**S7 Fig. Schematic representation of the parcellation scheme of the VOTC.** Anatomical regions were defined in each individual hemisphere according to major anatomical landmarks [95]. The ventral temporal sulci (collateral sulcus, occipito-temporal sulcus, and midfusiform sulcus) serve as medial/lateral borders of regions, whereas 2 coronal reference planes containing anatomical landmarks (posterior tip of the hippocampus and anterior tip of the parieto-occipital sulcus) serve as an anterior/posterior boundary for each region. Importantly, the posterior tip of the hippocampus separated the PTL and the ATL, and therefore contacts located anteriorly to the posterior tip of the hippocampus were labeled in the ATL. The anatomical location of each significant contact was determined in the individual brain according to this anatomical subdivision. The schematic locations of these anatomical structures are shown on a reconstructed cortical surface of the Colin27 brain. AntFG, anterior fusiform gyrus; antMTG/ITG, anterior part of the inferior and middle temporal gyri; antPHG, anterior segment of the parahippocampal gyrus; ATL, anterior temporal lobe; CoS, collateral sulcus; HIP, hippocampus; latFG, lateral part of the fusiform gyrus; medFG, medial part of the fusiform gyrus; MFS, midfusiform sulcus; MTG/ITG, inferior and middle temporal gyri; OCC, occipital lobe; OTS, occipito-temporal sulcus; PHG, parahippocampal gyrus; POS, parieto-occipital sulcus; PTL, posterior temporal lobe; VOTC, ventral occipito-temporal cortex.
(TIF)

**S8 Fig. Anatomical location and SEEG responses of the significant contacts in the PTL.** Anatomical location and electrophysiological responses of the 8 significant contacts in the PTL in 3 participants (P2, P3, P4) and of their adjacent contacts. The significant contacts are highlighted in red. Significant base responses were determined in the same way as significant identity-oddball responses by (1) epoching the EEG frequency spectrum into segments

centered on the first 6 base harmonics (i.e., 4 Hz, 8 Hz, etc.); (2) summing the amplitude values of these 6 frequency spectrum segments; and (3) transforming it into a Z-score (difference between the amplitude at the base frequency bin and the mean amplitude of the 22 surrounding bins, divided by the standard deviation of amplitudes in the corresponding 22 surrounding bins). Data underlying this figure are deposited on a Dryad repository: https://doi.org/10.5061/dryad.m8t391m. EEG, electroencephalography; SEEG, stereo electroencephalography; PTL, posterior temporal lobe.
(TIF)

**S9 Fig. Illustration of the identification procedure of significant intracerebral contacts from the raw FFT spectrum.** (A) The anatomical location of the contact that is illustrated in this example is shown in both coronal and sagittal views and indicated by a green dot. (B) Intracerebral frequency-domain responses recorded in this individual contact in the Face Name condition. Significant responses were determined by first segmenting the EEG frequency spectrum into 6 segments centered at the identity-oddball frequency and its harmonics. These 6 segments are illustrated by 6 gray bars on the $x$ axis and correspond to the length of each frequency spectrum segment. (C) Pattern of response of the 6 individual frequency segments is shown as gray lines. These segments were then summed, resulting in the green spectrum. The 0 mark represents the identity-change frequency. (D) Z-score transformation of the summed FFT spectrum. Z-score was computed as the difference between the amplitude at the identity-change frequency and the mean amplitude of the 22 surrounding bins, divided by the standard deviation of the 22 surrounding bins. The dashed line indicates the threshold of 3.1 ($p < 0.001$) that was used in intracerebral EEG to detect significant responses. (E) Computation of the subtraction between the Face Name condition and the sum of the two control conditions: Face Only and Name Only. First, raw FFT spectra of the Face Only and Name Only conditions were summed. This sum was then subtracted from the raw FFT spectrum of the Face Name condition. (F) Intracerebral frequency-domain responses resulting from the subtraction [Face Name − (Face Only + Name Only)]. Significant responses in this subtraction were determined by first segmenting the EEG frequency spectrum into 6 segments centered at the identity-oddball frequency and its harmonics. These 6 segments are illustrated by 6 gray bars on the $x$ axis that correspond to the length of each segment. (G) Pattern of response of the 6 individual segments is shown as gray lines. Because these responses resulted from the subtraction of two conditions (Face Only and Name Only) from one condition (Face Name), only the identity-oddball amplitude remains positive, while the amplitude of the surrounding bins is mostly negative. These segments were then summed, resulting in the green spectrum. As a consequence of the subtraction process, only the frequency of interest has a positive amplitude. The 0 mark represents the identity-change frequency. (H) Z-score transformation of the summed FFT spectrum. The dashed line indicates the threshold of 1.65 ($p < 0.05$) that was used in intracerebral EEG to detect significant responses at the subtraction [Face Name − (Face Only + Name Only)]. In this example, the Z-score at the identity-oddball frequency exceeds 1.65, indicating that the face-name response is not only explained by the sum of modality-specific responses. Data underlying this figure are deposited on a Dryad repository: https://doi.org/10.5061/dryad.m8t391m. EEG, electroencephalography; FFT, fast Fourier transform; FN, Face Name; FO, Face Only; NO, Name Only.
(TIF)

## Author Contributions

**Conceptualization:** Angélique Volfart, Bruno Rossion.

**Data curation:** Angélique Volfart, Jacques Jonas, Louis Maillard, Sophie Colnat-Coulbois.

**Formal analysis:** Angélique Volfart.

**Funding acquisition:** Bruno Rossion.

**Investigation:** Angélique Volfart, Jacques Jonas.

**Methodology:** Angélique Volfart, Bruno Rossion.

**Project administration:** Jacques Jonas, Bruno Rossion.

**Resources:** Bruno Rossion.

**Supervision:** Jacques Jonas, Bruno Rossion.

**Validation:** Bruno Rossion.

**Visualization:** Angélique Volfart, Jacques Jonas, Bruno Rossion.

**Writing – original draft:** Angélique Volfart, Jacques Jonas, Bruno Rossion.

**Writing – review & editing:** Angélique Volfart, Jacques Jonas, Louis Maillard, Sophie Colnat-Coulbois, Bruno Rossion.

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
