## [Editor Report · Decision Letter 0]

13 Aug 2019

Dear Dr Rossion, 

Thank you for submitting your manuscript entitled "Neurophysiological evidence for crossmodal person-identity representation in the human left ventral temporal cortex" for consideration as a Research Article by PLOS Biology.

Your manuscript has now been evaluated by the PLOS Biology editorial staff, as well as by an Academic Editor with relevant expertise, and I am writing to let you know that we would like to send your submission out for external peer review. Please accept my apologies for the delay in sending this decision to you.

However, before we can send your manuscript to reviewers, we need you to clarify, within your manuscript, some points regarding the implantation and use of depth electrodes in human patients:

1. Please mention whether the written consent was informed.

2. Please mention if your study adhered to the Helsinki Protocol or any other ethical guidelines

3. Please add the ID number of the approved protocol by the Ethics committee of the University Hospital of Nancy.

4. Please include a justification for the clinical use of depth electrodes, rather than grid electrodes, describing the advantages, disadvantages, and risks associated with them, and if any protective measures or other considerations were made in the procedure. Please add references.

5. Please explicitly mention of the measures that you took to track and minimize tissue damage.

6. Please submit as 'Related Files' a document in which the text changes are tracked.

In addition, we need you to complete your submission by providing the metadata that is required for full assessment. To this end, please login to Editorial Manager where you will find the paper in the 'Submissions Needing Revisions' folder on your homepage. Please click 'Revise Submission' from the Action Links and complete all additional questions in the submission questionnaire.

*Please be aware that, due to the voluntary nature of our reviewers and academic editors, manuscripts may be subject to delays during the holiday season. Thank you for your patience.*

Please re-submit your manuscript within two working days, i.e. by Aug 15 2019 11:59PM.

Kind regards,

Gabriel Gasque, Ph.D.,

Senior Editor

PLOS Biology

---

## [Decision Letter · Decision Letter 1]

9 Oct 2019

Dear Bruno,

Thank you very much for submitting your manuscript "Neurophysiological evidence for crossmodal person-identity representation in the human left ventral temporal cortex" for consideration as a Research Article at PLOS Biology. Your manuscript has been evaluated by the PLOS Biology editors, by an Academic Editor with relevant expertise, and by four independent reviewers. Please accept my apologies for the delay in sending the decision below to you.

In light of the reviews (below), we will not be able to accept the current version of the manuscript, but we would welcome resubmission of a much-revised version that takes into account the reviewers' comments. We cannot make any decision about publication until we have seen the revised manuscript and your response to the reviewers' comments. Your revised manuscript is also likely to be sent for further evaluation by the reviewers.

Your revisions should address the specific points made by each reviewer. Having discussed them with the Academic Editor, we think you should pay special attention to those raised by reviewer 2. 

Please submit a file detailing your responses to the editorial requests and a point-by-point response to all of the reviewers' comments that indicates the changes you have made to the manuscript. In addition to a clean copy of the manuscript, please upload a 'track-changes' version of your manuscript that specifies the edits made. This should be uploaded as a "Related" file type. You should also cite any additional relevant literature that has been published since the original submission and mention any additional citations in your response. 

Before you revise your manuscript, please review the following PLOS policy and formatting requirements checklist PDF: http://journals.plos.org/plosbiology/s/file?id=9411/plos-biology-formatting-checklist.pdf. It is helpful if you format your revision according to our requirements - should your paper subsequently be accepted, this will save time at the acceptance stage.

Please note that as a condition of publication PLOS' data policy (http://journals.plos.org/plosbiology/s/data-availability) requires that you make available all data used to draw the conclusions arrived at in your manuscript. If you have not already done so, you must include any data used in your manuscript either in appropriate repositories, within the body of the manuscript, or as supporting information (N.B. this includes any numerical values that were used to generate graphs, histograms etc.). For an example see here: http://www.plosbiology.org/article/info%3Adoi%2F10.1371%2Fjournal.pbio.1001908#s5.

For manuscripts submitted on or after 1st July 2019, we require the original, uncropped and minimally adjusted images supporting all blot and gel results reported in an article's figures or Supporting Information files. We will require these files before a manuscript can be accepted so please prepare them now, if you have not already uploaded them. Please carefully read our guidelines for how to prepare and upload this data: https://journals.plos.org/plosbiology/s/figures#loc-blot-and-gel-reporting-requirements.

Upon resubmission, the editors will assess your revision and if the editors and Academic Editor feel that the revised manuscript remains appropriate for the journal, we will send the manuscript for re-review. We aim to consult the same Academic Editor and reviewers for revised manuscripts but may consult others if needed.

We expect to receive your revised manuscript within two months. Please email us (plosbiology@plos.org) to discuss this if you have any questions or concerns, or would like to request an extension. At this stage, your manuscript remains formally under active consideration at our journal; please notify us by email if you do not wish to submit a revision and instead wish to pursue publication elsewhere, so that we may end consideration of the manuscript at PLOS Biology.

When you are ready to submit a revised version of your manuscript, please go to https://www.editorialmanager.com/pbiology/ and log in as an Author. Click the link labelled 'Submissions Needing Revision' where you will find your submission record. 

Sincerely,

Gabriel Gasque, Ph.D., 

Senior Editor

PLOS Biology

Reviewer remarks:

Reviewer #1: The authors present three studies of crossmodal integration in face-name processing, using scalp and intracranial EEG. They address theoretical models of crossmodal integration by comparing Face-Name pairings with Face or Name only stimulation, with the aim of providing evidence for presence of crossmodal person specific representations that are activated regardless of the modality of stimulation. The manuscript is well written, well presented and I have no major concerns about the methodology, however I must make it clear that I have no experience of intracranial EEG and cannot comment accurately on the reliability of these results. My minor comments and suggestions are:

Abstract: 

Change "names of a famous identity" to "names of a single famous identity"

Introduction:

P3 The sentence at the end suggests that verbal information is processed in the right ATL, is this intended?

P4 I don't know what is meant by "indifferent" neural responses

Methods:

P9 Putin is spelled Poutine, which as far as I know is a Canadian meal of chips and gravy....!

P12 The description of the chunking approach for the visualisation of oddball responses made no sense until I saw the data, it might be helpful to either simplify the text or include a simple diagram to explain this and perhaps other elements of the different oddball response quantification for the non EEG/non FPVS readers.

Figure 1: For the purposes of the figure perhaps keep the font simpler and just use white or black text, it is quite hard to read as it is.

Results:

P23 The quantification of the presence of significant oddball responses at the individual subject level is not described in enough detail. How was the Z score calculated? While most participants in the Supp Info figure show a clear peak the bottom right participant looks borderline. 

P24: A frequency plot, similar to Figure 2 in Experiment 1, is definitely needed for Experiment 2.

P27: Given that there was effectively no oddball response to the Name Only condition, it them seems odd to compare the ROIs of this non-response. I'm not sure this adds much. 

P29: I think Figure 7 and 6 would be better if they were combined, they are presenting identical types of analyses, just for different ROIs. I also think it would be useful to plot the right ROI. A more general point is does the whole scalp analysis add anything? Being able to show effects at the whole scalp level is great if the effects are strong enough that they can be shown in this way, but if they are not I'm not sure it adds much.

P29: I don't think the paired t-tests add anything beyond the ROI analysis and this section could be cut with no detriment to the paper. An alternative to this combination of scalp and ROI analyses and separate t-tests would be to use cluster permutation for everything.

P30: Did the three patients showing no response show no response at any contact at all? 

P31: I'm not clear what is meant by the number of sequences viewed by the participants or the mention of the experiment being repeated?

Discussion:

Without right ATL implants the specificity of the left ATL to this crossmodal representation is unknown but the authors are very clear about this and mention it multiple times.

Reviewer #2: To date, it is not known how a specific face and a name identity are integrated at a neural level into a common (cross-modal) representation. This is an open and fundamental question in neuroscience, of interest to both basic and clinical research. The authors discuss the importance of the ventral anterior temporal lobes (vATLs) in this process. To address this question and the involvement of the ATLs, the authors developed a cross-modal frequency-tagging paradigm, coupled with brain activity recordings via scalp and intracerebral electroencephalography (EEG). They run two independent experiments with scalp EEG and one with intracerebral EEG, and they conclude that their results point towards a key role of the left vATL in the automatic retrieval of face-name identity representations. 

This is a well thought and interesting study; however, the manuscript lacks important considerations of some potential confounds and limitations of the methods. In addition, the scalp EEG results do not provide strong information about responses in the ATLs but they only provide information about occipito-temporal regions. The intracerebral EEG experiment lacks details to determine if results are significant or not. The following issues should be addressed to strengthen this interesting study and make it suitable of publication:

Major comments (essential to support current conclusions)

- The authors should explain the theory behind frequency-tagging better, cite the first studies using this method (D. Regan, etc), and discuss why they chose to use it. They state in one paragraph in the introduction that they have developed an “original” cross-modal frequency-tagging paradigm. What is original about their method? Other studies have used frequency-tagging to explore cross-modal links (e.g. Colon et al., 2015). What are the advantages over other methods to study cross-modal interactions? E.g. why they do not assess cross-modal interactions measuring ERPs? 

- If the objective of the study was to explore the integration of face-name representations, why didn’t the authors design the paradigm with two different frequencies to tag faces and names? This would have given them an indicator of nonlinear interaction, that manifests itself in the form of responses at “intermodulation” frequencies (e.g., f1 ± f2, see Zemon & Ratliff, 1984; Appelbaum et al, 2008; Giani et al., 2012, Boremanse et al., 2013). 

- What is the spatial location of the authors’ ROIs in a circular view (looking down at the top of the head) topography? Why do the authors consider the source of these signals is in the ATL? They even state that face-name responses were found over posterior sites (second paragraph in Results, Experiment 1). They also do not compare the spatial localization of face-name responses with that of base frequency responses. Thus, how spatially specific are the face-name responses to the ATLs?

- In Figure 7B, the circular topography shows an area of strong modulation in the amplitude at an anterior right location, maybe in the frontal lobe, or maybe in a fronto-temporal contact. They do not report significant differences here, but it might be because they only report individual paired t-tests in the occipito-temporal regions. Is this amplitude difference significant and in which contact is located? 

- The authors should show that the SNR is all ROIs for the scalp EEG and also in all contacts for the intracranial EEG to determine that the effects are not due to better SNR in some contacts than others. Also, for experiment 2, the authors should show if there are differences between hemispheres for the base frequency responses. Otherwise it is unclear whether the differences they see in the Face-name condition and sum of control conditions between hemispheres are due to differences in SNR between hemispheres. 

- For the intracranial EEG recordings, the methods need to be written and shown in a clearer way. Given that intracranial EEG gives better spatial information than scalp EEG, the authors should show the contacts’ location for each patient in a supplementary figure. Especially important to support the conclusions drawn in this manuscript, the authors should show how anterior these contacts are within the temporal lobes. 

- The authors find that 6 out of the 196 contacts in the pooled intracranial recordings show significant responses at the frequency of person identity change. These 6 contacts belong to 4 patients. Where exactly are all contacts located and what do the contacts surrounding these 6 contacts show? What contacts show base frequency responses?

- The contact of patient 2 in figure 8 looks very posterior. Why do the authors consider this ATL? They should show clearly in a figure the spatial limits of what they consider ATL. 

- The authors state that “strikingly” three of 196 contacts show a “pure” face-name association response. I am not sure how striking is this. What is the likelihood of having 3 out of 196 contacts in 7 patients giving this response? I do not believe that a strong conclusion can be drawn from this about the role of ATL with this result. There is no control region to compare: maybe 3 out of 196 contacts in pure visual cortex would also give this result? 

Minor comments

- Figure 2 upper panel is not clear. For some frequencies there’s only a blue bar, for some others a red bar, and for others a combination of both. There is no explanation in the legend about this. 

- In the third paragraph of the introduction there is a typo:

“On the one hand, verbal and non-verbal person-specific information may be processed separately in the right and left anterior temporal lobe (ATL), respectively” 

In the following sentence they state this correctly: left ATL has been proposed to be part of verbal person-specific information, while right ATL has been proposed to differentially contribute to processing faces. 

- Abbreviations FSVP, SEEG need explanation where they first appear

- All figures could benefit from a more detailed legend.

Reviewer #3: In normal healthy and epileptic subjects, authors tested the competing hypotheses that verbal and non-verbal person-specific information may be processed separately in the right and left anterior temporal lobe, respectively (Hypothesis 1) or if there is a semantic ‘hub’ in the bilateral anterior temporal lobes, integrating modality-specific representations into a shared amodal/transmodal representation (Hypothesis 2). In lengthy experiments (whose design and method of analysis are extremely hard to follow) they found "neural" responses that were modality specific and selective to person's identity in the posterior sites (category specific responses are known in this region) but then they also found a modality non-specific semantic "concept" hub in the anterior temporal lobe (also very famously known with single unit recordings in human subjects (as referenced by authors). Sadly, they conclude that this is left lateralized without providing recordings directly from the right hemisphere. They argue that the lack of right intracerebral recordings was due to scalp EEG findings suggesting unilateral responses--- but lack of scalp findings does not justify lack of recordings. In other words, the paper does not seem to advance any theories and or settle the competition between the two hypotheses. 

Unlike single unit or gamma band recordings in the brain, I have hard time understanding what "neural" signal this 0.5Hz represents. What is so neural about it? The electrophysiological responses were analyzed at the frequency of oddball presentation 0.571 Hz and its harmonics. Interestingly in Face-Name task this oddball has a more robust structure whereas in the other two tasks the "conceptual" oddball is too randomly and too frequently presented perhaps presenting a confounding effect here. As authors anticipated "conceptual oddball" responses appeared in that signal spectrum exactly at the frequency of person identity change (4Hz/7 = 0.5713 Hz), and one might argue that Exp 2 and 3 did not provide a solid structure for testing the same oddball effect. Only Task 1 did. Also I have a problem with including 6 different colors for names and yet no specific color change for faces. Also face images subtended about 6.78° x 5.47° of visual angle, but the name condition subtended 13.65° x 1.62° of visual angle.

Reviewer #4: This is a sophisticated and interesting study. FPVS is emerging as a powerful cognitive neuroscience tool and this paper provides important developments from face recognition research into semantic knowledge. It is nice to see studies that combine investigations of healthy participants (with EEG) and neurological patients (SEEG) – who provide unique opportunities to localise the evoked responses.

The study is well conducted and thoroughly analysed. I only had one or two minor queries about the method. I had a few thoughts about interpretation and the siting of the current study in the broader literature about semantic representation and the anterior temporal regions. These can easily be dealt with in revision. In general, I think it would be good to consider data beyond those in the current study when considering the role of left and right ATL in semantic representation. There is now a considerable body of research using multiple methods and computational models. This includes at least one existing study of knowledge of people probed from their pictured faces and spoken names which is consistent with the current results. This work has also considered how semantic function might show bilateral yet asymmetries across different types of stimuli and input.

Specific points:

Abstract

It might be useful to note that semantic information (about people and any other concept) is more than just the association of faces and names. A key central aspect of semantic concepts is the ability to generalise knowledge across all modalities and over time. Thus there are many forms of knowledge, including names, that are linked to each person.

Like the main text, the abstract discusses left vATL functions with an implicit message that the results are different for other regions. Whilst the study can make some excellent observations about the left vATL, because the right vATL was not sampled then little can be said about it. Note – the broader semantics literature does have a consideration of left and right vATL functions and how they might work together.

Introduction

p.4: I think that it is important to note that the difference between left and right SD patients is graded with a _relative_ difference between words and faces. It is never an absolute difference and not a classical double dissociation. The same graded differences are observed in TLE resection patients and in functional neuroimaging (see various papers by Rice et al. 2015; 2017)

p.4-5: important to note that the hub-and-spoke model provides a much broader framework for semantic memory in general including knowledge about people. The same model shows that activating semantic information to the specific level is demanding and thus is more vulnerable to damage (see Rogers et al 2004; Rogers & Patterson 2007; Rogers et al 2015) 

p.4-5: In terms of a lack of evidence about integrated representations for verbal and visual forms of people: (a) if we assume that this is part of general semantics then there is converging evidence for multimodal semantic representations in semantic dementia, fMRI, TMS, computational models, etc. (for an overview, see Lambon Ralph et al NRN 2017) – with, for example, Visser et al (2011) showing that the same vATL region as observed in the SEEG in this study is activated by pictures, sounds and spokjen names; (b) SD patients show especially poor performance for semantic knowledge probed at the specific level including people and show strong item associations when knowledge for a concept is probed using words or pictures (indicating degradation of a shared representation); (c) at least one fMRI study has probed people knowledge with pictures and words (Rice et al 2017 Proc Roy Soc) which found bilateral vATL regions that responded to both the spoken names and pictures of familiar people, landmarks and animals.

Methods:

Could you say something about how the baseline frequency was determined. I believe in previous face-only FPVS experiments, a higher frequency was used. I assume that finding the correct frequency is critical.

Discussion

If I have understood correctly then only the left vATL was sampled in the SEEG? And scalp EEG cannot be used for localising responses in these anterior regions. I think it is important to make it clear, therefore, that we cannot consider what the right vATL might be doing from these data. Parallel work from semantic dementia, fMRI, rTMS and subdural grids suggests a bilateral division of function perhaps with some graded differences that align with connectivity variations (see Rice et al., for reviews and meta-analyses). And combined rTMS-fMRI suggests that the division of labour across left and right can change in line with demand (see Binney et al 2015; Jung et al 2016). Accordingly, the current study gives us interesting insights about the left vATL but much less about other regions.

In this context on p.34, I do not understand the statement “Such a unilateral face-name locus was unexpected considering that proponents of the unitary “hub” theory postulate that both ATL contribute to and are activated during multimodal semantic processing”. As noted by the authors in the previous sentence the right vATL was not probed with SEEG and thus this statement rests on the scalp distributions from EEG. Past work has already show that (a) laterality differences seems to be much stronger in posterior than anterior temporal regions (see Rice et al. reviews and meta-analysis, and Hoffman et al Cortex 2018); (b) the anterior temporal areas are both engaged by verbal and nonverbal stimuli (e.g., Visser et al) and (c) that these bilateral involvements are asymmetric for any tasks that require speech production of written word reading (see Rice et al 2015 meta-analysis). There are various computational models of the hub-and-spoke model that show how these asymmetric bilateral involvements can follow from differences in functional/structural connectivity (e.g., Lambon Ralph et al 2001; Shapiro et al 2013).

p.34-35: please see comments above for the Introduction about at least one previous study that has found a common (bilateral) vATL involvement in both faces and names at the specific level (Rice et al Proc Roy Society).

---

## [Decision Letter · Decision Letter 2]

22 Jan 2020

Dear Dr Rossion,

Thank you for submitting your revised Research Article entitled "Neurophysiological evidence for crossmodal (face-name) person-identity representation in the human left ventral temporal cortex" for publication in PLOS Biology. I have now obtained advice from the original reviewers 1, 2, and 3, and have discussed their comments with the Academic Editor. You will note that reviewer 1, George Stothart, has signed his comments. 

Based on the reviews, we are positive about your study. However, we want to you address, with textual modifications, the lingering concerns expressed by reviewer 2 before we can make a decision about publication. Please also make sure to address the data and other policy-related requests noted at the end of this email.

We expect to receive your revised manuscript within two weeks. Your revisions should address the specific points made by reviewer 2. Please submit the following files along with your revised manuscript:

In addition to the remaining revisions and before we will be able to formally accept your manuscript and consider it "in press", we also need to ensure that your article conforms to our guidelines. A member of our team will be in touch shortly with a set of requests. As we can't proceed until these requirements are met, your swift response will help prevent delays to publication.

*Copyediting*

*Published Peer Review History*

*Early Version*

*Submitting Your Revision*

Sincerely,

Gabriel Gasque, Ph.D., 

Senior Editor

PLOS Biology

DATA POLICY:

-- Please also ensure that the figure legends in your manuscript include information on where the underlying data can be found, and ensure your supplemental data file/s has a legend.

Reviewer remarks:

Reviewer #1, George Stothart: My comments were relatively minor and mostly suggested removing redundant analyses, which the authors have done to my satisfaction. 

Reviewer #2: Although I find this study interesting, and the authors addressed each of my points, I do not think that any strong conclusions can be made about the role of the anterior temporal lobes in cross-modal processing. For this manuscript to be published, the conclusions drawn about the role of the left ATL in the integration of face and name information, and the last sentence of the abstract must be changed in order to reflect the limitations that the study has. 

Scalp EEG results point towards a general occipito-temporal role in the integration of verbal and non-verbal person identity-specific information. After reading the responses to my and other reviewers' comments, and all the changes the authors made, I agree that this is a strong result. However, I still do not see how the intracranial EEG support these results. First, recordings are restricted in each patient only to the left hemisphere. They do not find any pure face-name response on the PTL contacts (out of 41) but they find 3 contacts in the ATL (out of 196) that show a pure face-name response. Again, I do not see how this can be statistically significant. What is the likelihood of having 3 significant contacts out of 196 vs. 0 out of 41 contacts by chance? In addition, how having 3 out of 7 patients showing this result is a high proportion? I believe the way intracranial EEG was applied as a method is not suitable to respond the questions the authors posited in the introduction. 

Finally, the authors should be careful when talking about neural populations, since they do not test this directly. Even if an area acts as a crossmodal hub, unless the activity of single neurons is analyzed, nothing can be concluded about the neural populations per se. Even with the intracranial EEG, the frequency tagging could reflect the existence of two different neural populations within the vicinity of the contact, one responding to faces and one to names. Unless single units are recorded and analyzed no conclusion can be made about the existence of a crossmodal population of neurons (this should also be modified in the abstract as well as in the main body of the manuscript).

Reviewer #3: I thank the authors for their replies.

---

## [Editor Report · Decision Letter 3]

9 Mar 2020

Dear Dr Rossion,

On behalf of my colleagues and the Academic Editor, Winrich A. Freiwald, I am pleased to inform you that we will be delighted to publish your Research Article in PLOS Biology. 

Early Version

PRESS 

Kind regards,

Vita Usova

Publication Assistant, 

PLOS Biology

on behalf of

Gabriel Gasque,

Senior Editor

PLOS Biology